# Neural Atoms: Propagating Long-range Interaction in Molecular Graphs through Efficient Communication Channel

**Xuan Li**[1*]   **Zhanke Zhou**[1*]   **Jiangchao Yao**[2,3]   **Yu Rong**[4]   **Lu Zhang**[1]   **Bo Han**[1†]

[1]Hong Kong Baptist University   [2]CMIC, Shanghai Jiao Tong University
[3]Shanghai AI Laboratory   [4]Tencent AI Lab

{csxuanli, cszkzhou, ericluzhang, bhanml}@comp.hkbu.edu.hk
sunarker@sjtu.edu.cn   yu.rong@hotmail.com

## Abstract

Graph Neural Networks (GNNs) have been widely adopted for drug discovery with molecular graphs. Nevertheless, current GNNs mainly excel in leveraging short-range interactions (SRI) but struggle to capture long-range interactions (LRI), both of which are crucial for determining molecular properties. To tackle this issue, we propose a method to abstract the collective information of atomic groups into a few *Neural Atoms* by implicitly projecting the atoms of a molecular. Specifically, we explicitly exchange the information among neural atoms and project them back to the atoms' representations as an enhancement. With this mechanism, neural atoms establish the communication channels among distant nodes, effectively reducing the interaction scope of arbitrary node pairs into a single hop. To provide an inspection of our method from a physical perspective, we reveal its connection to the traditional LRI calculation method, Ewald Summation. The Neural Atom can enhance GNNs to capture LRI by approximating the potential LRI of the molecular. We conduct extensive experiments on four long-range graph benchmarks, covering graph-level and link-level tasks on molecular graphs. We achieve up to a 27.32% and 38.27% improvement in the 2D and 3D scenarios, respectively. Empirically, our method can be equipped with an arbitrary GNN to help capture LRI. Code and datasets are publicly available in https://github.com/tmlr-group/NeuralAtom.

## 1 Introduction

Graph neural networks (GNNs) show promising ability of modeling complex and irregular interactions (Wu et al., 2020; Chen et al., 2021; Thomas et al., 2023; Zhang et al., 2023b). In particular, GNNs attract growing interest in accelerating drug discovery due to the accurate prediction of the molecular properties (Wieder et al., 2020; Li et al., 2022; Zhang et al., 2022c; Ma et al., 2021; 2022).

The basic elements of a molecule are different types of atoms, and the interactions among atoms can be generally categorized into short-range interactions (SRI) and long-range interactions (LRI), as illustrated in Fig. 1. Specifically, SRI, such as covalent bonds or ionic bonds, acts over relatively *small distances* that are typically within the range of a few atomic diameters. By contrast, LRI, such as hydrogen bonds and coulomb interactions, operates over *much larger distances* compared to the atomic diameter. Despite the weaker strength compared to SRI, LRI plays a significant role in determining both the physical and chemical properties of molecules (Harvey, 1989; Darden et al., 1993; Sagui & Darden, 1999; Winterbach et al., 2013; Leeson & Young, 2015; Zhang et al., 2022a).

In Fig. 1, we give an illustration of SRI and LRI in a molecular graph. The SRIs here, *e.g.*, the covalent bond between carbon (●) and hydrogen (○), are denoted as solid lines ( ●—○ ), which involves the electron exchange of atoms to establish the stabilization of the molecular structure. As

---

*Equal contributions.
†Correspondence to Bo Han (bhanml@comp.hkbu.edu.hk)

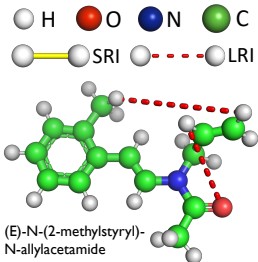

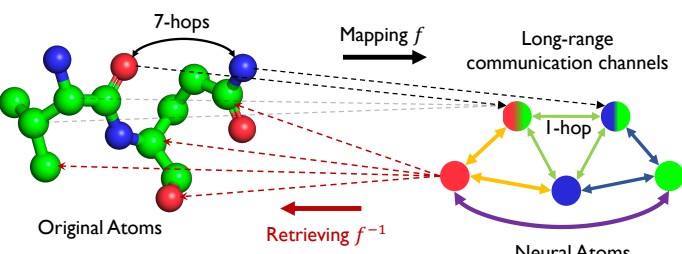

Figure 1: An exemplar molecular with the long-range interactions (dash lines) and short-range interactions (solid lines).

Figure 2: Illustration of Neural Atoms. The mapping function $f$ is to project the original atoms to Neural Atoms, and the retrieving function $f^{-1}$ aims to inject back the information, allowing the GNN to capture LRI via the interaction between Neural Atoms.

such, the SRIs are considered to be the explicit, one-hop edges of the molecular graph. On the other hand, the LRIs (denoted as dashed lines) are in the form of the implicit force depending on the distance between atoms in the 3D range and the molecular structure and could involve atoms with multiple hops of distance. For example, the hydrogen bond (◯, ●) can involve the five-hop-length path (◯, ●, ●, ●, ●, ●) with four middle atoms, whereas the hydrogen bond (◯, ◯) can even involve the ten-hop-length path ( ◯, ●, ●, ●, ●, ●, ●, ●, ●, ●, ●, ◯ ) with nine middle atoms.

By aggregating information from neighborhoods in each layer, GNNs can effectively capture the SRI. However, they are intrinsically ineffective in capturing the LRI that is located among distant nodes. Simply stacking more layers to perceive distant atoms will encounter exponentially increasing neighbor nodes. Inevitably, excessive spurious information is compressed, regardless of its correlation to the target atom. Such a phenomenon would result in the well-known over-squashing problem (Alon & Yahav, 2020; Topping et al., 2022) and over-smoothing problem (Rong et al., 2020b).

In this work, we aim to design an effective and efficient mechanism to enhance GNNs for capturing long-range interactions. We notice that in areas of computational chemistry and molecular modeling, the pseudo atoms can be employed to group atoms into a more manageable and computationally efficient representation. These pseudo atoms are not real atoms with physical properties but are introduced for simplification and convenience in calculations. One can approximate the effects of LRIs by introducing pseudo atoms strategically, making simulations of complex molecular systems more tractable while still retaining the essential characteristics of these interactions. However, pseudo atoms cannot be directly applied here, as they are typically manually designed with expert knowledge.

In this work, we propose to project all the original atoms into a few neural atoms to abstract the collective information of atomic groups in a molecule. We explicitly exchange information among neural atoms and project it back to the atoms' representations as an enhancement. Under this mechanism, the neural atoms establish communication channels among distant nodes, which can reduce the interaction scope of arbitrary node pairs into a single hop. Note that the proposed method is architecture-agnostic and computation-efficient as an enhancement for capturing LRI.

Empirically, we evaluate our method on three long-range graph benckmarks (Vijay et al., 2022b) with both link-level and graph-level tasks, with a focus on 2D intra-molecular interaction without the use of 3D coordinate information. We achieve up to a 27.32% improvement in the Peptides-struct dataset with different kinds of commonly used GNNs. In addition to the 2D intra-molecular interaction, we also evaluate the effectiveness of neural atom under 3D inter-molecular scenarios in the OE62 dataset for three common backbones, following the setting of Ewald-based message passing (Kosmala et al., 2023). With fewer parameters and the absence of 3D information, neural atom achieves competitive performance *w.r.t.* the Ewald-based approach. Our main contributions are as follows:

- We formalize the concept of neural atoms, and instantiate it with dual self-attention mechanisms to learn the atoms-neural-atoms projection and model the interactions among neural atoms (Sec. 3).
- We conduct extensive experiments on long-range molecular datasets for property prediction and structure reconstruction, and empirically justify that our method can boost various GNNs (Sec. 4).
- We provide an in-depth understanding of our method from a physical perspective and reveal its intrinsic connection with a commonly-used LRI calculation method, Ewald Summation (Sec. 5).

## 2 PRELIMINARIES

**Notation.** An undirected graph is denoted as $\mathcal{G} = (\mathcal{V}, \mathcal{E})$, where $\mathcal{V}$ is the node set and $\mathcal{E}$ is the edge set. For each node $v \in \mathcal{V}$, its $D$-dimension node feature is denoted as $x_v \in \mathbb{R}^D$. Besides, $\boldsymbol{h}_v^{(\ell)}$ denote the node representation of node $v$ in $\ell$-th layer for a $L$-layer GNNs, where $0 \leq \ell \leq L$ and $\boldsymbol{h}_v^{(0)} = x_v$.

**Graph Neural Networks.** Given a graph $\mathcal{G}$, the neighbor information aggregate function $f^{(\ell)}$ and the node representation update function $\phi^{(\ell)}$, we can formulate the $\ell$-th layer operation of GNNs as

$$\boldsymbol{h}_v^{(\ell)} = \phi^{(\ell)}\big(\boldsymbol{h}_v^{(\ell-1)}, f^{(\ell)}(\boldsymbol{h}_v^{(\ell-1)})\big),$$

where $\mathcal{N}(v) = \{u \in \mathcal{V} | (u, v) \in \mathcal{E}\}$ are the neighbor nodes of node $v$. Different GNNs vary from the design of the function $f^{(\ell)}$ and $\phi^{(\ell)}$. For example, GCN (Kipf & Welling, 2016) define its $f^{(\ell)}$ as

$$f^{(\ell)}(\boldsymbol{h}_v^{(\ell-1)}) = \Sigma_{u \in \mathcal{N}(v) \cup \{v\}} 1/\sqrt{\hat{d}_u \hat{d}_v} \mathbf{W}^{(\ell)} \boldsymbol{h}_u^{(\ell-1)},$$

and use `ReLu` function as $\phi^{(\ell)}$, where $\mathbf{W}$ is the learnable parameter for filtering the graph signal. Likewise, GIN (Xu et al., 2018) obtains neighbor information by $f^{(\ell)}(\boldsymbol{h}_v^{(\ell-1)}) = \Sigma_{u \in \mathcal{N}(v) \cup \{v\}} \boldsymbol{h}_u^{(\ell-1)}$, and update node representation via a feed-forward network. In addition, the node representations in the last GNN layer would then be transformed by a feed-forward network for specific downstream tasks. For example, for graph-level classification tasks, the whole representations $\boldsymbol{H} \in \mathbb{R}^{N \times d}$ would be transformed to the logits $\hat{\boldsymbol{Y}} \in \mathbb{R}^C$ with $C$ classes that usually through the graph pooling operation.

**Multi-head Attention Mechanism.** Consider a molecular graph with $N$ nodes, the input of the attention mechanism $f_{\text{Att}}$ consists of three components, termed as query $\boldsymbol{Q} \in \mathbb{R}^{k \times d_k}$, key $\boldsymbol{K} \in \mathbb{R}^{N \times d_k}$, and value $\boldsymbol{V} \in \mathbb{R}^{N \times d_v}$, where $d_k$ and $d_v$ are dimensions. As such, the $f_{\text{Att}} = \sigma(\boldsymbol{Q}\boldsymbol{K}^\top)V$ is calculated via the dot product of the query and the key, with $\sigma$ denoting the activation function. To extend the attention to the multi-head case, we further utilize the linear projection for $\boldsymbol{Q}, \boldsymbol{K}$, and $\boldsymbol{V}$ to yield $M$ representation subspace. The multi-head attention mechanism allows us to learn different attentions between query and key, thus enhancing the model expressivity. Namely, with $M$ attention heads,

$$\text{MultiHead}(\boldsymbol{Q}, \boldsymbol{K}, \boldsymbol{V}) = \boldsymbol{W}^O \|_{m=1}^M \boldsymbol{O}_m, \quad \text{where} \quad \boldsymbol{O}_m = f_{\text{Att}}(\boldsymbol{Q}\boldsymbol{W}_m^Q, \boldsymbol{K}\boldsymbol{W}_m^K, \boldsymbol{V}\boldsymbol{W}_m^V),$$

where $\boldsymbol{W}^O \in \mathbb{R}^{Md_h \times d_o}$ is the linear projection matrix for learning the subspace representation of the contracted head outputs to output dimension $d_o$. Besides, $\boldsymbol{W}_m^Q \in \mathbb{R}^{d_k \times d_k}$, $\boldsymbol{W}_m^K \in \mathbb{R}^{d_k \times d_k}$, and $\boldsymbol{W}_m^V \in \mathbb{R}^{d_v \times d_v}$ are the projection matrixes in the $m$-th head for query, key, and value, respectively.

**Graph Hierarchical Learning.** An intuitive idea is to group the intermediate atoms and abstract their information into a single node. This could reduce the information-propagating length of LRI by decreasing the potential intermediate atoms, which helps enhance the distant atoms' information strength. Two direct implementations, virtual node (Gilmer et al., 2017) and graph pooling (Ying et al., 2018; Ma et al., 2019; Ekagra et al., 2020; Baek et al., 2021) can be employed to capture the potential LRI. However, these two strategies have inherent drawbacks. The virtual node might make a model that cannot distinguish the necessary information, as it aggregates information via simple add-up. Graph pooling, on the other hand, could make the GNN unable to propagate and utilize the LRI information. A detailed discussion can be found in Appendix C and D.5.

**Ewald Summation.** As a calculation algorithm for molecular dynamics simulation, Ewald Summation (Toukmaji & Board Jr., 1996) calculates the energy of the interaction by processing the atoms' position and their charges via mathematical models. The Ewald Summation decomposes the actual interaction into the short-range and long-range parts, where the former decays vastly in real space and the latter only exhibits fast decay in frequency space. The short-range part can be calculated by employing a summation with the distance cut-off. Meanwhile, the long-range part, transformed with the Fourier transformation, can be calculated via a low-frequency cut-off summation.

## 3 METHOD: BUILDING EFFICIENT LONG-RANGE COMMUNICATION CHANNEL THROUGH NEURAL ATOMS

In this section, we first introduce the concept of neural atoms and how it can benefit GNNs to capture LRI (Sec. 3.1). Next, we elaborate on the attention-based approach to realize neural atoms (Sec. 3.2). Finally, we connect it with the Ewald Summation from the graph representation learning (Sec. 3.3).

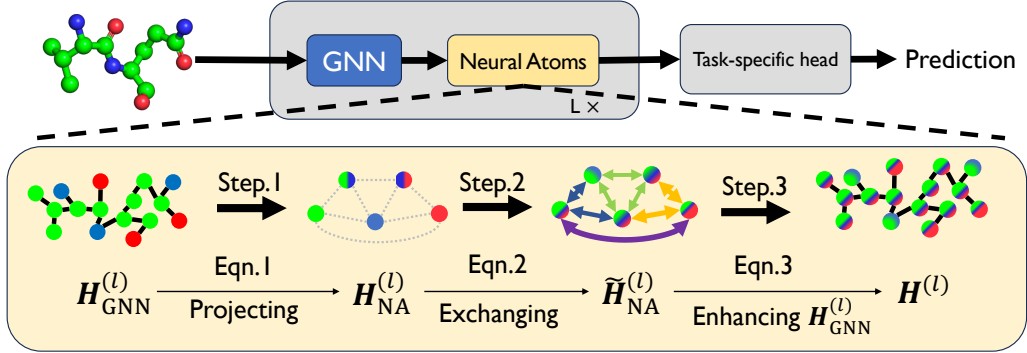

Figure 3: The proposed Neural Atom framework aims to obtain graph representation for different downstream tasks. The Neural Atom can enhance arbitrary by injecting LRI information via the interaction of neural atoms. We demonstrate the information exchange by the mixture of colors.

## 3.1 FORMALIZING THE NEURAL ATOMS

We propose the neural atoms inspired by the pseudo atoms of molecular dynamics simulations, which abstract prefixed atom groups into individual representations. We formalize neural atoms as follows.

**Definition 1.** *Neural atoms encompass a collection of virtual, parameterized atoms that symbolize a cluster of atoms within a designated molecular graph. The process entails the acquisition of knowledge that enables the transformation of conventional atoms into neural atoms, along with their interactions. This transformation can be technically executed with model-agnostic methodologies.*

Based on the definition 1, we analyze the three major advantages of neural atoms as follows.

**Advantage 1** (Learnable projection from atoms to neural atoms). Different from the pseudo atoms that require expert knowledge of specific domains, the neural atoms are designed to extract the LRI automatically w.r.t. the current molecular graph dataset. The number of neural atoms is flexible and decoupled from the size of the original molecular graph. Different neural atoms are expected to gather data from distinct local regions to preserve local information to the greatest extent possible.

**Advantage 2** (Reducing multi-hop long-range interaction to single-hop). Interactions among atoms that are widely separated are transformed into interactions among neural atoms, effectively narrowing the interaction range from any arbitrary pair of atoms to a single step. Namely, long-range information from distant atoms can be effectively communicated through interactions among neural atoms.

**Advantage 3** (GNN-agnostic and plug-in-and-play). Employing layer-wise collaboration with GNNs, the long-range information captured by neural atoms can be backward projected to the original atoms, thus enhancing the short-range information captured by the GNNs' message passing.

However, it is non-trivial to achieve an effective and efficient implementation of Def. 1. Therein, the technical designs of (1) the learnable projection, (2) the interaction among neral atoms, and (3) retrieving neural atoms' LRI information to enhance original atom embeddings need to be determined. In what follows, we solve these technical problems and elaborate on the implementation of neural atoms via the attention mechanism and the collaboration with GNNs.

## 3.2 IMPLEMENTING THE NEURAL ATOMS

In this part, we introduce the technical details of implementing neural atoms. The overall inference pipeline of GNN with neural atoms is illustrated in Fig. 3. Briefly, we obtain the atom representations $H_{\text{GNN}}^{(\ell)}$ via the $\ell$-th GNN model and then apply neural atoms to get the enhanced representations $H^{(\ell)}$. Therein, the three steps are: (1) project atom representations $H_{\text{GNN}}^{(\ell)}$ to neural atom representations $H_{\text{NA}}^{(\ell)}$, (2) exchange information among neural atoms to get $\tilde{H}_{\text{NA}}^{(\ell)}$, and (3) backward project the neural atoms and enhance the atom representations as $(H_{\text{GNN}}^{(\ell)}, \tilde{H}_{\text{NA}}^{(\ell)}) \mapsto H^{(\ell)}$. The details are as follows.

**Step-1. Project atom representations $H_{\text{GNN}}^{(\ell)}$ to neural atom representations $H_{\text{NA}}^{(\ell)}$.** To achieve this projection (Advantage 1), we commence by initializing learnable weights $Q_{\text{NA}}^{(\ell)} \in \mathbb{R}^{K \times d_k}$, where

$K$ ($\ll N$) specifies the number of neural atoms as a hyperparameter and $d_k$ is the embedding dimension. By employing the multi-head attention (MultiHead), we can learn the grouping function for obtaining neural atoms representing the intermediate atoms. Wherein, the atom representations $\boldsymbol{H}_{\text{GNN}}^{(\ell)} \in \mathbb{R}^{N \times d}$ are mapped to the key $\boldsymbol{K} \in \mathbb{R}^{N \times d_h}$ and value $\boldsymbol{V} \in \mathbb{R}^{N \times d_h}$ by linear projection weights $\boldsymbol{W}_K \in \mathbb{R}^{d_h \times d_h}$ and $\boldsymbol{W}_V \in \mathbb{R}^{d_h \times d_h}$, respectively. The allocation matrix $\hat{\boldsymbol{A}}_m$ for $m$-th head is obtained as $\hat{\boldsymbol{A}}_m = \sigma(\boldsymbol{Q}_{\text{NA}}^{(\ell)} \boldsymbol{K}^\top) \in \mathbb{R}^{K \times N}$, and $\oplus$ denotes operation for combining representations, $e.g.$, sum or a feed forward network. The representations $\boldsymbol{H}_{\text{NA}}^{(\ell)}$ of neural atoms are obtained by:

$$\boldsymbol{H}_{\text{NA}}^{(\ell)} = \text{LayerNorm}\left(\boldsymbol{Q}_{\text{NA}}^{(\ell)} \oplus \text{MultiHead}(\boldsymbol{Q}_{\text{NA}}^{(\ell)}, \boldsymbol{H}_{\text{GNN}}^{(\ell)}, \boldsymbol{H}_{\text{GNN}}^{(\ell)})\right), \tag{1}$$

where the LayerNorm represents the operation of Layer Normalization (Ba et al., 2016).

**Step-2. Communicate information among neural atoms $\boldsymbol{H}_{\text{NA}}^{(\ell)} \mapsto \tilde{\boldsymbol{H}}_{\text{NA}}^{(\ell)}$.** Then, we explicitly exchange information among neural atoms to capture the long-range interactions in this molecular graph, as in Advantage 2. We further employ the self-attention mechanism for efficient information exchange as:

$$\tilde{\boldsymbol{H}}_{\text{NA}}^{(\ell)} = \text{LayerNorm}\left(\boldsymbol{H}_{\text{NA}}^{(\ell)} \oplus \text{MultiHead}(\boldsymbol{H}_{\text{NA}}^{(\ell)}, \boldsymbol{H}_{\text{NA}}^{(\ell)}, \boldsymbol{H}_{\text{NA}}^{(\ell)})\right). \tag{2}$$

**Step-3. Project neural atoms back and enhance the atoms' representation $(\boldsymbol{H}_{\text{GNN}}^{(\ell)}, \tilde{\boldsymbol{H}}_{\text{NA}}^{(\ell)}) \mapsto \boldsymbol{H}^{(\ell)}$.** So far, the obtained $\tilde{\boldsymbol{H}}_{\text{NA}}^{(\ell)}$ contains information from different atom groups. To cooperate the $\tilde{\boldsymbol{H}}_{\text{NA}}^{(\ell)}$ with the original molecular atom, we aggregate the allocation matrix $\hat{\boldsymbol{A}}_m$ of different heads and project the neural atoms into the atom space with size $N$. Here, we perform the matrix reduction operations ($e.g.$, mean or summation) to aggregate the multi-head $\hat{\boldsymbol{A}}_m$ and get $\tilde{\boldsymbol{A}}_{\text{NA}}^{(\ell)}$, which allows the model to learn different allocation weights. As in Advantage 3, the final representations $\boldsymbol{H}^{(\ell)}$ are obtained by enhancing atom representations $\boldsymbol{H}_{\text{GNN}}^{(\ell)}$ with neural atom representations $\tilde{\boldsymbol{A}}_{\text{NA}}^{(\ell)} \tilde{\boldsymbol{H}}_{\text{NA}}^{(\ell)}$, $i.e.$,

$$\boldsymbol{H}^{(\ell)} = \boldsymbol{H}_{\text{GNN}}^{(\ell)} \oplus \tilde{\boldsymbol{A}}_{\text{NA}}^{(\ell)} \tilde{\boldsymbol{H}}_{\text{NA}}^{(\ell)}, \quad \text{s.t.} \quad \tilde{\boldsymbol{A}}_{\text{NA}}^{(\ell)} = \text{Aggregate}\left(\{\hat{\boldsymbol{A}}_m\}_{m=1}^M\right)^\top \in \mathbb{R}^{N \times K}. \tag{3}$$

**The overall procedure.** We summarize the forward pipeline in Algorithm 1. In brief, given the atom representations of a molecular graph $\boldsymbol{H}_{\text{GNN}}^{(\ell-1)}$ in $(\ell-1)$-th layer, we first get the updated atoms' representations $\boldsymbol{H}_{\text{GNN}}^{(\ell)}$ by the GNN in $\ell$-th layer to capture the SRI. Then, the atoms $\boldsymbol{H}_{\text{GNN}}^{(\ell)}$ are projected to neural atoms $\tilde{\boldsymbol{A}}_{\text{NA}}^{(\ell)} \tilde{\boldsymbol{H}}_{\text{NA}}^{(\ell)}$ to capture the LRI with the three above steps. Finally, the enhanced representations $\boldsymbol{H}^{(\ell)}$ are obtained by mixing both atoms' and neural atoms' representations.

**Remark 1** (Expressiveness)**.** We map the atoms in the molecular graph to $K$ neural atoms, which are far smaller than the size of the atoms in the original molecular graph. This allows us to identify the relevance of information since the attention mechanism aggregates the information according to the similarity between the embedding of neural atoms and the original atoms.

In addition to the expressiveness, neural atoms also enjoy the advantages of low computational complexity and the ability to scale to large molecular graphs. The running time comparison of neural atoms and other approaches can be found in the Appendix. B.

**Remark 2** (Complexity and Scalability)**.** Instead of directly modeling the atom interaction via a fully connected graph (Wu et al., 2023; Ladislav et al., 2022), we map the potential interaction into the space constructed by neural atoms, which is more sparse compared to the original graph. For scalability, one can apply neural atoms to large molecular graphs by employing a linear Transformer-like Performer (Choromanski et al., 2021) or BigBrid (Zaheer et al., 2020). To reduce the computation, one can prune the connections between neural atoms and original atoms based on the attention score.

## 3.3 CONNECTION TO THE EWALD SUMMATION

Here, we utilize the aforementioned Ewald sum matrix (Faber et al., 2015) to show and understand the interaction among particles. The Ewald sum matrix consists of pair-wise atomic interaction strength modeled by Ewald summation. Specifically, the Ewald summation decomposes the interaction into SRI (denoted as $x_{ij}^{(r)}$) and LRI (termed as $x_{ij}^{(\ell)}$). The SRI can be calculated by summation in real space, while the LRI requires to be transformed into reciprocal space by Fourier transform, $i.e.$,

---

**Algorithm 1** Message propagation with neural atoms.

---

**Require:** Molecular graph $\mathcal{G}$, atoms feature $X$, and GNN model $f$.

1: Initialize $\boldsymbol{H}^{(0)} \leftarrow X$
2: **for** $\ell = 1 \ldots L$ **do**
3:    $\boldsymbol{H}_{\text{GNN}}^{(\ell)} \leftarrow f^{(\ell)}(\boldsymbol{H}^{(\ell-1)}, \mathcal{G})$                    ▷ Obtaining SRI information.
4:    $\boldsymbol{H}_{\text{NA}}^{(\ell)} \leftarrow \text{LayerNorm}\left(\boldsymbol{Q}_{\text{NA}}^{(\ell)} \oplus \text{MultiHead}(\boldsymbol{Q}_{\text{NA}}^{(\ell)}, \boldsymbol{H}_{\text{GNN}}^{(\ell)}, \boldsymbol{H}_{\text{GNN}}^{(\ell)})\right)$ ▷ Project $\boldsymbol{H}_{\text{GNN}}^{(\ell)}$ to $\boldsymbol{H}_{\text{NA}}^{(\ell)}$ via Eqn. 1.
5:    $\tilde{\boldsymbol{H}}_{\text{NA}}^{(\ell)} \leftarrow \text{LayerNorm}\left(\boldsymbol{H}_{\text{NA}}^{(\ell)} \oplus \text{MultiHead}(\boldsymbol{H}_{\text{NA}}^{(\ell)}, \boldsymbol{H}_{\text{NA}}^{(\ell)}, \boldsymbol{H}_{\text{NA}}^{(\ell)})\right)$ ▷ Information exchanging via Eqn. 2.
6:    $\tilde{\boldsymbol{A}}_{\text{NA}}^{(\ell)} \leftarrow \text{Aggregate}\left(\{\hat{\boldsymbol{A}}_m\}_{m=1}^M\right)^\top$                    ▷ Aggregate allocation matrix $\hat{\boldsymbol{A}}_m$.
7:    $\boldsymbol{H}^{(\ell)} \leftarrow \boldsymbol{H}_{\text{GNN}}^{(\ell)} \oplus \tilde{\boldsymbol{A}}_{\text{NA}}^{(\ell)} \tilde{\boldsymbol{H}}_{\text{NA}}^{(\ell)},$                    ▷ Project $\boldsymbol{H}_{\text{NA}}^{(\ell)}$ back to enhance $\boldsymbol{H}_{\text{GNN}}^{(\ell)}$ via Eqn. 3.
8: **end for**
9: **return** $\boldsymbol{H}^{(L)}$.                    ▷ Final atom embeddings for the task-specific head.

---

$$x_{ij} = Z_i Z_j \underbrace{\sum_{\mathbf{L}} \frac{\text{erfc}\left(a \left\|\mathbf{r}_i - \mathbf{r}_j + \mathbf{L}\right\|_2\right)}{\left\|\mathbf{r}_i - \mathbf{r}_j + \mathbf{L}\right\|_2}}_{\text{SRI: } x_{ij}^{(r)}} + \underbrace{\frac{Z_i Z_j}{\pi V} \sum_{\mathbf{G}} \frac{e^{-\|\mathbf{G}\|_2^2/(2a)^2}}{\|\mathbf{G}\|_2^2} \cos\left(\mathbf{G} \cdot (\mathbf{r}_i - \mathbf{r}_j)\right)}_{\text{LRI: } x_{ij}^{(\ell)}} + x_{ij}^{(s)}.$$

The above equation gives the interatomic interaction strength, *i.e.*, the non-diagonal elements in the Ewald sum matrix. The $Z_i$ and $\mathbf{r}_i$ are the atomic number and position of the $i-$th atom, and $V$ is the unit cell volume. Besides, $L$ denotes the lattice vectors within the distance cutoff, $G$ is the non-zero reciprocal lattice vectors, and $a$ is the hyperparameter that controls the summation converge speed.

The strength of SRI and LRI are determined by the number of atomic positive charges. Their difference lies in the manner of calculating the distance between atoms. The $x_{ij}^{(r)}$, corresponding to the strength of the interaction in the real space, is calculated by the error function $\text{erfc}$ and the Euclidian distance $|\mathbf{r}_i - \mathbf{r}_j + \mathbf{L}|_2$ between atoms. The $x_{ij}^{(\ell)}$ describe the interaction strength in the reciprocal space, which is calculated by the distance given by the dot product of $G$ and the interatomic distance $(\mathbf{r}_i - \mathbf{r}_j)$. Whereas the self-energy $x_{ij}^{(s)}$ is a constant term that is irrelevant to the distance.

**Remark 3.** Recall in Eqn. 3, the final atom representations $\boldsymbol{H}^{(\ell)}$ are obtained by combining both $\boldsymbol{H}_{\text{GNN}}^{(\ell)}$ and $\tilde{\boldsymbol{A}}_{\text{NA}}^{(\ell)} \tilde{\boldsymbol{H}}_{\text{NA}}^{(\ell)}$. Associating with the Ewald summation, the $\boldsymbol{H}_{\text{GNN}}^{(\ell)}$ contains the information of the SRI and self-energy term, while the enhanced $\tilde{\boldsymbol{A}}_{\text{NA}}^{(\ell)} \tilde{\boldsymbol{H}}_{\text{NA}}^{(\ell)}$ is to approximate the LRI term.

## 4 EXPERIMENTS

In this section, we empirically evaluate the proposed method on real-world molecular graph datasets for both graph-level and link-level tasks. All the datasets require modeling LRI for accurate prediction on downstream tasks. We aim to provide answers to the following two questions. **Q1**: How effective are the proposed methods on real-world molecular datasets with common GNNs? **Q2**: How does the grouping strategy of neural atoms affect the performance of different GNNs for capturing the LRI?

**Setup.** For 2D intra-molecule interaction, we employ the molecular datasets (Peptides-Func, Petides-Struct, PCQM-Contact) that exhibit LRI from Long Range Graph Benchmarks (LRGB) (Vijay et al., 2022b). We implement the neural atom with GNNs in the GraphGPS framework (Ladislav et al., 2022), which provides various choices of positional and structural encodings. For the 3D inter-molecule scenario, we employ the OE62 (Stuke et al., 2020) with long-ranged London dispersion interaction for evaluation, which has over 60,000 organic molecules with more than 100 atoms for each molecule. All the experiments are run on an NVIDIA RTX 3090 GPU with AMD Ryzen 3960X CPU. Detailed experiment settings and dataset description are shown in the Appendix A.

**Baseline.** For 2D intra-molecule interaction, We employ the GCN (Kipf & Welling, 2016), GINE (Hu et al., 2020b), GCNII (Chen et al., 2020) and GatedGCN (Bresson & Laurent, 2017) and GatedGCN augmented augmented with Random Walk Structure Encoding (RWSE) as the baseline GNNs. To capture the LRI, one could adopt the fully connected graph to obtain interatomic interaction for distant atom pairs explicitly. Here, we adopt the Graph Transformers for comparison. Specifically, we introduce the fully connected Trasnforemr (Vaswani et al., 2017) with Laplacian Positon Encodings (Vijay et al., 2022a), SAN (Kreuzer et al., 2021) and the recent GraphGPS (Ladislav et al., 2022),

Table 1: Test performance on three LRGB datasets. Shown is the mean $\pm$ s.d. of 4 runs.

| Model | Peptides-func | Peptides-struct | PCQM-Contact |
|---|---|---|---|
| | AP $\uparrow$ | MAE $\downarrow$ | MRR $\uparrow$ |
| Transformer+LapPE | $0.6326 \pm 0.0126$ | $0.2529 \pm 0.0016$ | $0.3174 \pm 0.0020$ |
| SAN+LapPE | $0.6384 \pm 0.0121$ | $0.2683 \pm 0.0043$ | $0.3350 \pm 0.0003$ |
| GraphGPS | $0.6535 \pm 0.0041$ | $0.2500 \pm 0.0005$ | $0.3337 \pm 0.0006$ |
| GCN | $0.5930 \pm 0.0023$ | $0.3496 \pm 0.0013$ | $0.2329 \pm 0.0009$ |
| **+ Neural Atoms** | $\mathbf{0.6220 \pm 0.0046}$ | $\mathbf{0.2606 \pm 0.0027}$ | $\mathbf{0.2534 \pm 0.0200}$ |
| GINE | $0.5498 \pm 0.0079$ | $0.3547 \pm 0.0045$ | $0.3180 \pm 0.0027$ |
| **+ Neural Atoms** | $\mathbf{0.6154 \pm 0.0157}$ | $\mathbf{0.2553 \pm 0.0005}$ | $\mathbf{0.3126 \pm 0.0021}$ |
| GCNII | $0.5543 \pm 0.0078$ | $0.3471 \pm 0.0010$ | $0.3161 \pm 0.0004$ |
| **+ Neural Atoms** | $\mathbf{0.5996 \pm 0.0033}$ | $\mathbf{0.2563 \pm 0.0020}$ | $\mathbf{0.3049 \pm 0.0006}$ |
| GatedGCN | $0.5864 \pm 0.0077$ | $0.3420 \pm 0.0013$ | $0.3218 \pm 0.0011$ |
| **+ Neural Atoms** | $\mathbf{0.6562 \pm 0.0075}$ | $\mathbf{0.2585 \pm 0.0017}$ | $\mathbf{0.3258 \pm 0.0003}$ |
| GatedGCN+RWSE | $0.6069 \pm 0.0035$ | $0.3357 \pm 0.0006$ | $0.3242 \pm 0.0008$ |
| **+ Neural Atoms** | $\mathbf{0.6591 \pm 0.0050}$ | $\mathbf{0.2568 \pm 0.0005}$ | $\mathbf{0.3262 \pm 0.0010}$ |

Table 2: Validation energy MAE and MSE comparison on OE62 dataset.

| | Energy MAE $\downarrow$ | Energy MSE $\downarrow$ | Number of Params. |
|---|---|---|---|
| SchNet (Schütt et al., 2017) | 0.1351 | 0.0658 | 2.75 M |
| + Ewald Block | **0.0811** | **0.0301** | 12.21 M |
| **+ Neural Atoms** | **0.0834** | **0.0309** | 2.63 M |
| PaiNN (Schütt et al., 2021) | 0.6049 | 0.0133 | 12.52 M |
| + Ewald Block | **0.0590** | **0.0134** | 15.68 M |
| **+ Neural Atoms** | **0.0558** | **0.0122** | 6.05 M |
| DimeNet++ (Gasteiger et al., 2020) | 0.0501 | 0.0117 | 2.76 M |
| + Ewald Block | **0.0479** | **0.0107** | 4.75 M |
| **+ Neural Atoms** | 0.0551 | 0.0129 | 1.97 M |

which utilize GNNs and Transformer to capture short-range and long-range information respectively. For 3D inter-molecule scenario, we employ the SchNet (Schütt et al., 2017), PaiNN (Schütt et al., 2021) and DimeNet++ (Gasteiger et al., 2020) as the baseline GNNs. We implement the neural atom under the training framework from Ewald-based Message Passing (Kosmala et al., 2023) without the usage of 3D coordinate information. We cooperate our neural atom with different GNN models to evaluate its performance, denoted as *+ Neural Atoms*. For the 3D scenario, we compare neural atom with the SOTA Ewald-based method (Kosmala et al., 2023), denoted as *+ Ewald Block*.

## 4.1 QUANTITATIVE RESULTS

**2D intra-molecular interaction.** Shown as Tab. 1, all the GNNs achieve significant improvement with the assistance of neural atoms. The GNNs gain improvement from $8.17\%$ to $12.55\%$ on the peptides-func dataset. The GNNs also receive significant improvement on the peptides-struct dataset, at most $27.32\%$ for the GINE model. The improvement for different GNNs on various datasets empirically proves that neural atoms can enhance the common GNNs to capture LRI on 2D molecular graphs. Especially for the GatedGCN, which exceeds the Transformer with LapPE on the peptides-func dataset and shows competitive results for the other counterparts on both the peptides-func and PCQM-Contact datasets. We notice that more powerful GNNs, especially with edge learning or edge attention filtering mechanisms, could lead to better performance. Compared to other GNN models, the significant improvement of GatedGCN could be the incoming information filtering via the gate mechanism, which allows the neural atoms to obtain representation with rich SRI information.

**3D inter-molecular interaction.** We directly adopt the neural atoms to the 3D scenario without 3D coordinate information. Note that for all baseline GNNs, we only take **10** neural atoms and half the embedding dimension compared to the *Ewald Block*. Shown as Tab. 2, our method achieves competitive performance even with the absence of 3D coordinate information for different GNNs with only half the parameters, showing the generalization and effectiveness of neural atoms. Especially for the PaiNN, with less than half the parameters of the Ewald-based approach, our neural atom achieves the best performance for both energy MAE and MSE metrics.

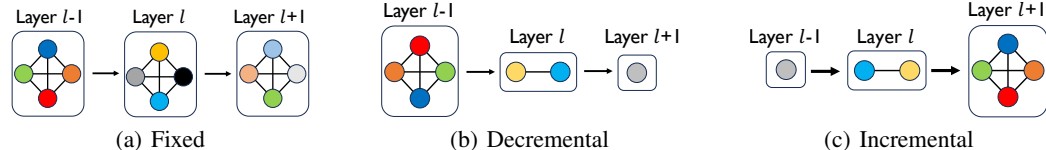

(a) Fixed        (b) Decremental        (c) Incremental

Figure 4: Neural Atom grouping strategies.

Table 3: Test performance for different grouping strategies on **Peptides-struct**.

| Model | Fixed | Incremental | Decremental |
|---|---|---|---|
| GCN | **0.2582 ± 0.0011** | 0.3239 ± 0.0014 | 0.2606± 0.0003 |
| GINE | **0.2559 ± 0.0001** | 0.2795 ± 0.0012 | 0.2578 ± 0.0017 |
| GCNII | 0.2579 ± 0.0025 | 0.4084 ± 0.0025 | **0.2563 ± 0.0020** |
| GatedGCN | 0.2592 ± 0.0017 | **0.2568 ± 0.0009** | **0.2569 ± 0.0007** |
| GatedGCN+RWSE | **0.2521 ± 0.0014** | 0.2600 ± 0.0012 | 0.2568± 0.0005 |

Table 4: Test performance for different proportions (#neural atoms / #atoms) on **Peptides-func**.

| Model | proportion = 0.1 | proportion = 0.5 | proportion = 0.9 |
|---|---|---|---|
| GCN | 0.5859 ± 0.0073 | 0.5903 ± 0.0054 | **0.6220 ± 0.0046** |
| GINE | 0.6128 ± 0.0060 | 0.6147 ± 0.0121 | **0.6154 ± 0.0157** |
| GCNII | 0.5862 ± 0.0066 | 0.5909 ± 0.0099 | **0.5996 ± 0.0033** |
| GatedGCN | 0.6533 ± 0.0030 | **0.6562 ± 0.0044** | 0.6562 ± 0.0075 |
| GatedGCN+RWSE | 0.6550 ± 0.0032 | 0.6565 ± 0.0074 | **0.6591 ± 0.0050** |

## 4.2 THE VARYING CHOICE OF $K$

As previously noted, our methodology involves the categorization of atoms inside the initial molecule into $K$ neural atoms. When determining the appropriate hyperparameter $K$, we take into account the fluctuating quantity of atoms present in the molecules, aligning it with the average number of atoms found within the dataset. We present an analysis of the impact of different strategies of the $K$. The technique involves setting the value of $K$ as a proportion to the average number of atoms on the dataset at the initial layer. The fixed technique, denoted as *fixed K*, involves setting the value of $K$ as a proportion relative to the average number of atoms in the dataset, as seen in Fig. 4(a). The *decremental*, where each layer is determined as a proportion to the previous value of $k$, as depicted in Fig. 4(b). The *incremental* strategy can be considered as the antithesis of the *decremental* approach, wherein the value of $K$ is increasing sequentially, as depicted in Fig. 4(c). We assess various approaches for determining the count of neural atoms on peptides-struct, as depicted in Tab. 3.

The *fixed* configuration enables the model to establish a stable interaction space, which is generated by a predetermined amount of neural atoms in each layer. This arrangement can be advantageous for most GNNs. While employing the *decremental* approach, the model is capable of adopting an interaction space characterized by an inverse pyramid structure, hence facilitating the acquisition of information from a local to a global perspective. The use of *incremental* methods may not be appropriate for the LRI scenario due to the potential disruption of the local structure. This disruption can occur when the atom representation is mapped to an increasingly complicated interaction space.

An experiment is conducted to investigate the effects of different proportions for the *decremental* approach, as presented in Tab. 4. Reduced grouping proportions are associated with heightened levels of assertive grouping techniques, while conversely, higher proportions tend to be linked to more moderate approaches. It has been observed that employing a more moderate grouping approach yields superior outcomes. The grouping procedure has the potential to coarsen the molecular graph, hence potentially improving the preservation of the local structure, specifically the SRI.

## 5 UNDERSTANDING

In order to demonstrate the impact of neural atoms on the process of grouping atoms and establishing high-level connections between them, as well as its capability to form meaningful groups of atoms, we adopt the Mutagenicity dataset (Morris et al., 2020) as a case study. This dataset offers explicit labels for atom groups within the molecular graph category, specifically accounting for the presence of $-NO$ and $-NH_2$ groups, which are indicative of the *mutagen* properties of the respective molecules.

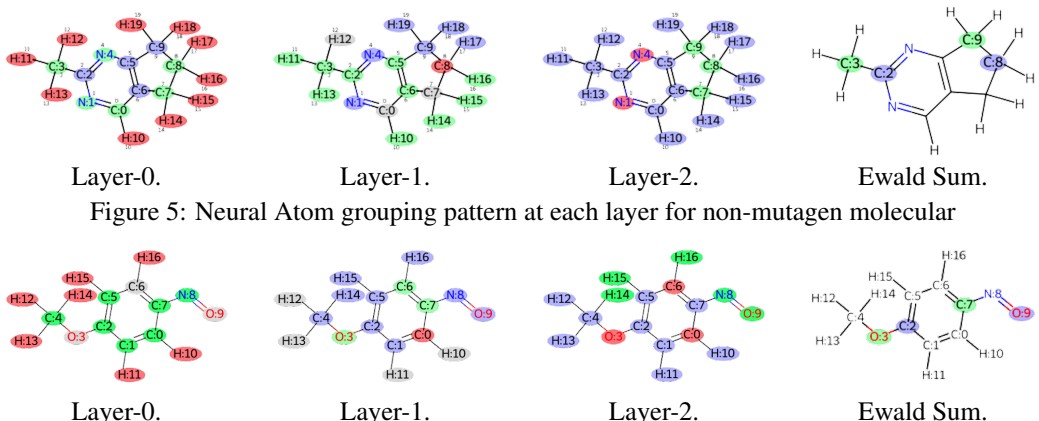

Figure 5: Neural Atom grouping pattern at each layer for non-mutagen molecular

Figure 6: Neural Atom grouping pattern at each layer for mutagen molecular ($-NO$)

To represent the potential interatomic interactions, we utilize the Ewald sum matrix to visualize. The Ewald sum matrix comprises elements that represent the Ewald Summation for distinct interatomic interactions, as denoted by the respective row and column indices within the matrix. In this study, we utilize a three-layer Graph Isomorphism Network (GIN) model (Xu et al., 2018), with a fixed number of neural atoms. Specifically, our model consists of four neural atoms in each layer. The atom allocation matrix is visualized through the assignment of a distinct color to each atom within the original chemical graph based on the index of the highest attention weight within the matrix. The allocation pattern for the neural atoms at each layer, as well as the interatomic interactions suggested by the Ewald sum matrix, are visualized in Fig. 5 to 6. The observed grouping pattern is consistent with the interatomic interaction as suggested by the Ewald sum matrix, with the application of thresholding for the purpose of enhancing visual clarity. Atoms displaying the same color are indicative of their possession of a high degree of interaction potential. As illustrated in Fig. 5, the atoms located within the range of (C:3-C:9) and (C:2-C:8) are assigned to separate neural atoms. This enables the model to depict their interaction by sharing information between these neural atoms.

Furthermore, we employ visual representation to depict the molecular structure of the mutagen compound containing the $-NO$ group, as illustrated in Fig. 6. In the primary layer, hydrogen atoms (H) and oxygen atoms (O) are segregated into distinct neural atoms, regardless of the multi-hop distance between them. This enables the model to get the atom representation corresponding to each element. Within the remaining layers, the constituents of $-NO$ (N:8 and O:9) are organized into a neural atom, enabling the model to capture their representation comprehensively. This facilitates the prediction of the molecular graph. It is worth noting that the observed grouping pattern is consistent with the interatomic interaction, as shown by the Ewlad sum matrix. The atoms denoted as (C:2-O:9) and (C:7-O:3) are assigned to distinct neural atoms. The model is capable of accurately representing the LRI even when there are multiple hops and intermediate atoms between the interacting entities. The presented visualization showcases the efficacy of our proposed methodology in facilitating the connection between remote atoms. Additionally, the atom groups identified in this study have the potential to significantly impact the accurate prediction of molecular graph features. We provide more visualization understanding of the attention pattern in Appendix H.

## 6 DISCUSSION AND CONCLUSION

**Extension.** One intuitive approach is extending the neural atom to leverage the atomic coordinate information better to capture the LRI. Another possible direction is to instill expert knowledge in the grouping strategy to improve the interpretation ability and discriminability of the grouped atoms.

**Conclusion.** In this study, we aim to enhance GNNs to better capture long-range interactions. We achieve this by transforming original atoms into neural atoms, facilitating information exchange, and then projecting the improved information back to atomic representations. This novel approach reduces interaction distances between nodes to a single hop. Extensive experiments on four long-range graph benchmarks validate our method's ability to enhance any GNN to capture long-range interactions.

## ACKNOWLEDGEMENT

XL, ZKZ, and BH were supported by the NSFC General Program No. 62376235, Guangdong Basic and Applied Basic Research Foundation Nos. 2022A1515011652 and 2024A1515012399, Tencent AI Lab Rhino-Bird Gift Fund, HKBU Faculty Niche Research Areas No. RC-FNRA-IG/22-23/SCI/04, and HKBU CSD Departmental Incentive Scheme. JCY was supported by the National Key R&D Program of China (No. 2022ZD0160703), 111 plan (No. BP0719010), and National Natural Science Foundation of China (No. 62306178). LZ was supported by Hong Kong Research Grant Council Early Career Scheme (HKBU 22201419). The authors thank Rong Tao for his advice and support in experiments.

## LIMITATION

This work mainly focuses on the molecular graph without 3D coordinate information, which could benefit the incorporation of LRI, as it depends on the interatomic distance in 3D space. In addition, our work pays more attention to the intra-molecular interaction than the inter-molecular interaction, where more valuable information might lie. For example, protein docking also involves the formation of the hydrogen bond between two biochemical components (Wu et al., 2012).

## ETHIC STATEMENT

This paper does not raise any ethical concerns. This study does not involve human subjects, practices to data set releases, potentially harmful insights, methodologies and applications, potential conflicts of interest and sponsorship, discrimination/bias/fairness concerns, privacy and security issues, legal compliance, and research integrity issues.

## REPRODUCIBILITY STATEMENT

The experimental setups for training and evaluation, as well as the hyperparameters, are described in detail in Section 4 and Appendix A, and the experiments are all conducted using public datasets.

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

# Appendix

## Table of Contents

## A    REPRODUCTION DETAILS

### A.1    HYPERPARAMETERS

**The 2D scenario.** We performed our experiment on four seeds and reported the mean with standard deviation as the final result. We summarize the common hyperparameters that are shared across different models on the LRGB datasets, along with the model-specific hyperparameters, shown as Tab. 5 to Tab. 8.

Table 5: Common hyperparameters for datasets from Long Range Graph Benchmark.

| Hyperparameter | PCQM-Contact | Peptides-func | Peptides-struct |
|---|---|---|---|
| Dropout | 0 | 0.12 | 0.2 |
| Allocation matrix Grouping | mean | mean | mean |
| Positional Encoding | LapPE-10 | LapPE-10 | LapPE-10 |
| PE dim | 16 | 16 | 20 |
| PE encoder | DeepSet | DeepSet | DeepSet |
| Batch size | 256 | 128 | 128 |
| Learning Rate | 0.001 | 0.0003 | 0.0003 |
| # Epochs | 200 | 200 | 200 |

Table 6: Model-specific hyperparameters for PCQM-Contact.

| Hyperparameter | # GNN Layers | Hidden dim | # Heads | proportion | # Neural Atoms |
|---|---|---|---|---|---|
| GCN | 5 | 300 | 1 | 0.9 | 27 |
| GCNII | 5 | 100 | 2 | 0.8 | 24 |
| GINE | 5 | 100 | 1 | 0.95 | 28 |
| GatedGCN | 8 | 72 | 1 | 0.5 | 15 |

Table 7: Model-specific hyperparameters for Peptides-func.

| Hyperparameter | # GNN Layers | Hidden dim | # Heads | proportion | # Neural Atoms |
|---|---|---|---|---|---|
| GCN | 5 | 155 | 1 | 0.15 | 22 |
| GCNII | 5 | 88 | 1 | 0.2 | 27 |
| GINE | 5 | 88 | 2 | 0.9 | 135 |
| GatedGCN | 5 | 88 | 1 | 0.5 | 75 |

Table 8: Model-specific hyperparameters for Peptides-struct.

| Hyperparameter | # GNN Layers | Hidden dim | # Heads | proportion | # Neural Atoms |
|---|---|---|---|---|---|
| GCN | 5 | 155 | 1 | 0.15 | 22 |
| GCNII | 5 | 88 | 1 | 0.2 | 30 |
| GINE | 5 | 88 | 2 | 0.9 | 135 |
| GatedGCN | 5 | 88 | 1 | 0.5 | 135 |

**The 3D scenario.** We evaluate the neural atom based on the implementation of Ewald-based Message-Passing (Kosmala et al., 2023). We keep most of the hyperparameters the same as the Ewald-based approach. We fixed the number of Neural Atom to **10** for each block. Detailed comparison of specific hyperparameters for methods adopted in Tab. 2 across different datasets is listed in Tab. 9, 10 and 11.

### A.2    DATASET DETAILS

**2D molecular graph datasets.** The statistical information of the datasets is shown as Tab. 12. Note that all three datasets consist of multiple graphs and are evaluated under the inductive setting, which means that the evaluating portion of the dataset differs from the training counterparts. For the PCQM-Contact dataset, the task is to predict whether the distant node pairs (with more than 5 hops away in a molecular graph) would be in contact with each other in the 3D space, *i.e.*, forming hydrogen bonds, which is the inductive link prediction task. The metric for the performance of the model is measured

Table 9: Hyperparameters for SchNet.

| | Hidden Channels | # Filters | # Interactions |
|---|---|---|---|
| Baseline | 512 | 256 | 4 |
| Ewald-block | 512 | 256 | 4 |
| Neural Atom | 256 | 128 | 2 |

Table 10: Hyperparameters for PaiNN.

| | Hidden Channels | # RBF |
|---|---|---|
| Baseline | 512 | 128 |
| Ewald-block | 512 | 128 |
| Neural Atom | 256 | 64 |

Table 11: Hyperparameters for DimeNet++.

| | Hidden Channels |
|---|---|
| Baseline | 256 |
| Ewald-block | 256 |
| Neural Atom | 128 |

Table 12: Dataset statistical information. All the datasets consist of molecular graphs with LRI.

| Dataset | Total Graphs | Total Nodes | Avg Nodes | Mean Deg. | Total Edges | Avg Edges | Avg Short.Path. | Avg Diameter |
|---|---|---|---|---|---|---|---|---|
| pcqm-contact | 529,434 | 15,955,687 | 30.14 | 2.03 | 32,341,644 | 61.09 | 4.63±0.63 | 9.86±1.79 |
| pepfunc | 15,535 | 2,344,859 | 150.94 | 2.04 | 4,773,974 | 307.30 | 20.89±9.79 | 56.99±28.72 |
| pepstruct | 15,535 | 2,344,859 | 150.94 | 2.04 | 4,773,974 | 307.30 | 20.89±9.79 | 56.99±28.72 |

Table 13: Dataset statistical information for larger graph.

| Dataset | Total Graphs | Total Nodes | Total Edges | Task Type | Task Metric |
|---|---|---|---|---|---|
| ogbn-arXiv (Hu et al., 2020a) | 1 | 169,343 | 1,166,243 | Node multi-class classification | Accuracy |
| Amazon Product-Computer (Shchur et al., 2018) | 1 | 13,752 | 491,7222 | Node multi-class classification | Accuracy |

by the Mean Reciprocal Rank (MRR). Both Peptides-func and Peptides-struct are constructed from the same source but serve different purposes. The Peptides-func is a multi-label graph classification dataset for evaluating the model's ability to capture molecular properties. We adopt the unweighted mean Average Precision (AP) as the metric. The Peptides-struct is a multi-label graph regression dataset based on the 3D structure of the peptides and uses Mean Absolute Error (MAE) as the metric.

**3D molecular graph datasets.** The OE62 contains 61,489 unique organic molecular graphs, each of which consists of up to 174 atoms with 16 different elements (H, Li, B, C, N, O, F, Si, P, S, Cl, As, SE, Br, Te, I). All molecular graphs are related in the gas phase with density-functional theory (DFT). The LRIs in OE62 are majorly long-ranged London dispersion interactions with a large spatial extent compared to the average atom size. Models are required to calculate the interaction energy (in eV), which is then evaluated by Energy Mean Absolute Error (MAE) and Energy Mean Square Error (MSE) *w.r.t.* to the ground truth computed by DFT.

# B    RUNNING TIME COMPAIRSON

We show the running time for different models in Tab. 14, training with the hyperparameters given as Tab. 5 to Tab. 8.

Table 14: Wall-clock run times. Average epoch time (average of 5 epochs, including validation performance evaluation) is shown for each model and dataset combination.

| avg. time / epoch | Peptides-func | Peptides-struct | PCQM-Contact |
|---|---|---|---|
| GCN | 2.6s | 2.5s | 56.9s |
| + Neural Atom | 5.5s | 4.9s | 65.1s |
| GINE | 2.6s | 2.6s | 56.7s |
| + Neural Atom | 4.8s | 4.2s | 66.8s |
| GCNII | 2.5s | 2.3s | 56.9s |
| + Neural Atom | 4.7s | 5.1s | 59.4s |
| GatedGCN | 3.3s | 3.2s | 56.5s |
| + Neural Atom | 6.1s | 5.5s | 61.6s |
| GatedGCN+RWSE | 3.4s | 4.1s | 59.4s |
| + Neural Atom | 6.4s | 5.2s | 65.0s |
| Transformer+LapPE | 6.4s | 6.2s | 59.2s |
| SAN+LapPE | 60s | 57.5s | 205s |
| GraphGPS | 6.5s | 6.5s | 61.5s |

Table 15: Wall-clock run times. Average epoch time (average of 5 epochs, including validation performance evaluation) is shown for each model and dataset combination.

| avg. time / epoch | ogbn-arXiv | Amazon Product-Computer |
|---|---|---|
| GCN | 0.5s | 0.2s |
| + Neural Atom | 2.1s | 0.2s |
| GINE | 0.4s | 0.2s |
| + Neural Atom | 2.1s | 0.2s |
| GCNII | 0.4s | 0.2s |
| + Neural Atom | 2.2s | 0.3s |
| GatedGCN | 0.6s | 0.2s |
| + Neural Atom | 2.3s | 0.2s |
| Transformer+LapPE | OOM | 0.4s |
| GraphGPS | OOM | 0.4s |

In order to showcase the effectiveness of our suggested neural atom, we have conducted calculations to determine the duration of each training epoch. The results are presented in Tab. 14. Although our method requires slightly more time, it exhibits more computational efficiency compared to the Transformer approach, particularly in the case of the SAN with LapPE. We also provide a comparison on larger graphs in Tab. 13 to further demonstrate the running time gaps between the Transformer-based method and our proposed Nerual Atoms, and running times are listed in Tab. 15.

## C  COMPARISON WITH VIRTUAL NODE

The pipeline of virtual node and neural atoms for obtaining global graph information can be described as three steps.

- Information aggregating: The information of atoms within the molecular graph is aggregated into either virtual node or multiple neural atoms with pair-wise connection.
- Interaction among node/atoms: The second step shows differences in interaction among node/atoms. The virtual node commonly exists alone, which means there is only one super node and thus lacks the ability to interact with others like neural atoms.
- Backward Projection: The final step shows differences in terms of the interaction among nodes/atoms. As the virtual node commonly exists alone, it thereby lacks the ability to interact with others like neural atoms.

We provide a detailed comparison in Tab. 16.

Table 16: Comparison with virtual node.

|  | **Super/Virtual Node** | **Neural Atoms** |
|---|---|---|
| #Atoms or #Nodes | 1 (In most cases, there is only one single virtual node for aggregating the global graph information, as increasing the number of which might not lead to performance improvement.) | $K$ (Our neural atoms can be defined as the proportion of the average number of atoms of the molecular graph dataset, which is significantly smaller than the original number of atoms. The more neural atoms, the better the performance.) |
| Information aggregating | The global pooling method, e.g., global sum/mean/max pooling, treats all the information from nodes the same, thus lacking diversity among different nodes in the graph. | The multi-head Attention mechanism allows aggregating information with different weights according to the similarity between specific neural atoms and the original atoms, which allows diversity among different atoms. |
| Interaction among virtual node/neural atoms | None. (Since the virtual node usually exists alone, it thus lacks the ability to interact with others.) | A fully connected graph with an attention mechanism to bridge the neural atoms for information exchange. This allows the information located in a different part of the graph, even with a large hop distance, to share information based on embedding similarities. |
| Backward projection | Direct element-wise adding. The information of the virtual node is directly added to the representation of each node, which fuses the information from the single virtual node, which might lead to the similarity among different atoms. | Weighted combination. Neural atoms can fuse the information within according to the similarity score between neural atoms and the atoms in the original molecular graph. Such a mechanism allows the model to obtain diversity representation for further purposes. |

# D FURTHER DISCUSSIONS

In this section, we provide a detailed discussion of graph hierarchical learning (Sec. D.1), graph Laplacian position encoding (Sec. D.2), trustworthy GNNs learning (Sec. D.3), graph transformer (Sec. D.4) and virtual node (Sec. D.5).

## D.1 GRAPH HIERARCHICAL LEARNING

The GNNs can only propagate information through edges formed by the SRI without additional feature augmentation (Liu et al., 2022b; Gasteiger et al., 2019). To capture LRI, the model would inevitably aggregate information on the numerous intermediate atoms from the source node to the target node. The overwhelming information could suppress the information from distant target nodes, thus degenerating the ability of the model to capture LRI. One straightforward implementation is appending a virtual node (Gilmer et al., 2017) to graph with connection with all the atoms to extract global information to improve the model's performance without 3D position information available. The virtual node technique has been proven to increase the expressiveness and reduce under-reaching issues (Hwang et al., 2022), and the ability to approximate the Graph Transformer (Cai et al., 2023). However, the virtual node differs from the neural atom in terms of grouping strategy and information exchange mechanism. A detailed discussion can be found in Appendix C and D.5. The grouping operation is achieved by graph pooling for abstracting the node representation while preserving the local structure information (Ying et al., 2018; Ma et al., 2019; Ekagra et al., 2020; Baek et al., 2021). Such an operator allows the model to obtain multi-scale graph-level representation, which implicitly enhances the model to capture LRI. However, graph pooling is designed to obtain global representation without considering cooperating with the LRI and SRI information. It makes the GNN model unable to propagate and utilize the LRI information.

## D.2 GRAPH LAPLACIAN POSITION ENCODING

The position encoding can benefit graph learning by instilling distinguishable information to the node features, such as local structure and Laplacian eigenvectors (Ladislav et al., 2022). Standard GNNs are known to be bounded by the 1-Weisfeiler-Leman test (1-WL), meaning they fail to distinguish non-isomorphic graphs with 1-hop message passing. Instilling graph position encoding allows GNNs to be more expressive than the 1-WL test, as each node is equipped with the distinguishable information (Kreuzer et al., 2021; Ladislav et al., 2022).

## D.3 TRUSTWORTHY GNNS LEARNING

**Out-of-Distribution for molecular graphs.** Out-of-distribution occurs with the distribution shift from the training to the testing environment, which hardly holds in practice (Hu et al., 2020a; Koh et al., 2021; Zhang et al., 2023a; Zhu et al., 2023; Wang et al., 2023b;a). The performance of GNNs could be seriously degraded due to the graph distribution shift, which is caused by environmental factors during data collection (Wang et al., 2023c; Ding et al., 2021). The performance consistency of GNNs crossing different environments with potential distribution shifts is important for the models' trustworthiness (Chen et al., 2023).

**Data noise and safety.** The SRI within molecular graphs for drug design is an important property as they form the basis of the medicines. Training GNNs on private data requires expert knowledge for annotation and numerous hyperparameters tunning (Zhang et al., 2022b; 2023c). Protecting the training data from stealing after deploying the model is crucial for the construction of trustworthy GNNs (Zhou et al., 2023b). In addition, the inevitable potential annotation noise during data collection and construction should also be considered when designing and deploying trustworthy GNNs (Zhou et al., 2023a).

## D.4 GRAPH TRANSFORMER

A common enhancement of GNNs is Graph Transformer (GT) (Kreuzer et al., 2021; Ladislav et al., 2022; Rong et al., 2020a; Min et al., 2022; Liu et al., 2023), which uses the self-attention mechanism to process information from neighbors that allows the capture of complex and long-range relationships between nodes. However, with self-attention, a node may attend to a large number of nodes with

no direct edge connection, where the excessive information can also bring difficulty in capturing meaningful LRI that are usually sparse, and it could even involve irrelevant information during the message aggregation and update process. In practice, only marginal improvements can GTs achieve compared with GNNs (Vijay et al., 2022b), while the self-attention on the entire molecular makes GTs much more computationally expensive than GNNs.

## D.5 VIRTUAL NODE

Molformer (Wu et al., 2023) combines molecular motifs and 3D geometry information by heterogeneous self-attention to create expressive molecular representations. The paper uses a virtual atom as a starting point to extract the graph representation for downstream graph-level tasks. The paper proposes attentive farthest point sampling for sampling atoms in 3D space not only according to their coordinates but also their attention score. However, the virtual atom they utilize does not participate in the message aggregation nor graph-level representation extraction, as they claim to "locate a virtual node in 3D space and build connections to existing vertices." As such, the potential long-range interaction they capture might be due to the atom pair-wise heterogeneous self-attention, which differs from our method, where the long-range interaction is captured by both the attention mechanism (step.1 in Fig. 3) and the interaction among the neural atoms (step.2 in Fig. 3).

Gilmer et al. (2017) firstly introduce the concept of Message Passing Neural Networks (MPNN) to develop a unified framework for predicting molecular properties. The paper introduces the "virtual node" as an argument for global information extraction. The virtual node, connected to all other nodes within the graph, acts as the global communication channel, enhancing the model's ability to capture long-range interactions and dependencies in molecular graphs. The authors experimented with a "master node" connected to all other nodes, serving as a global feature aggregator. This approach showed promise in improving the model's performance, especially in scenarios where spatial information, e.g., 3D coordination, is limited or absent.

Hwang et al. (2022) study the benefit of introducing single or multiple virtual nodes for link prediction tasks. Virtual node, traditionally thought to serve as aggregated representations of the entire graph, is connected to subsets of graph nodes based on either randomness or clustering mechanisms. Such methodology significantly increases the expressiveness and reduces under-reaching issues in MPNN. The study reveals that virtual nodes when strategically integrated, can provide stable performance improvements across various MPNN architectures and are particularly beneficial in dense graph scenarios. Their virtual node differs from our Neural Atom regarding the grouping strategy and the information-exchanging mechanism.

Cai et al. (2023) investigates the relationship between MPNN and Graph Transformers (GT) via the bridge of the virtual node. It demonstrates that MPNN augmented with virtual nodes can approximate the self-attention layer of GT. Under certain circumstances, the paper provides a construction for MPNN + VN with $O(1)$ width and $O(n)$ depth to approximate the self-attention layer in GTs. The paper provides valuable insight into understanding the theoretical capabilities of MPNN with virtual nodes in approximating GT. Compared to our neural atoms, we do not focus on establishing a theoretical connection between MPNN and GT. Instead, we are interested in leveraging the attention mechanism's ability to construct an interaction subspace constructed by the neural atoms. As such, the subspace acts as a communication channel to reduce interaction distances between nodes to a single hop.

# E  PERFORMANCE COMPARISON FOR NEURAL ATOMS AND VIRTUAL NODES

In this section, we show the difference between Nerual Atoms and virtual node, w.r.t. the performance by increasing their number. Specifically, we borrow the setting of Tab. 1 by aligning the number of neural atoms and virtual nodes and the backbone GNN they used. We employ the "VirtualNode" data transform from the PyG framework and set all virtual nodes connected for a fair comparison.

Table 17: Performance for virtual nodes (VNs) and neural atoms (NAs) in Peptide-Func, evaluated by AP (the higher, the better).

| Model | Method | #VNs /#NAs $= 5$ | #VNs /#NAs $= 15$ | #VNs /#NAs $= 75$ | #VNs /#NAs $= 135$ |
|---|---|---|---|---|---|
| GCN | VNs | 0.5566 | 0.5543 | 0.5568 | 0.5588 |
|  | NAs | **0.5962** | **0.5859** | **0.5903** | **0.6220** |
| GINE | VNs | 0.5437 | 0.5500 | 0.5426 | 0.5426 |
|  | NAs | **0.6107** | **0.6128** | **0.6147** | **0.6154** |
| GCNII | VNs | 0.5086 | 0.5106 | 0.5077 | 0.5083 |
|  | NAs | **0.6061** | **0.5862** | **0.5909** | **0.5996** |
| GatedGCN | VNs | 0.5810 | 0.5868 | 0.5761 | 0.5810 |
|  | NAs | **0.6660** | **0.6533** | **0.6562** | **0.6562** |

Table 18: Performance for virtual nodes (VNs) and neural atoms (NAs) in Peptide-Struct, evaluated by MAE (the lower, the better).

| Model | Method | #VNs /#NAs $= 5$ | #VNs /#NAs $= 15$ | #VNs /#NAs $= 75$ | #VNs /#NAs $= 135$ |
|---|---|---|---|---|---|
| GCN | VNs | 0.3499 | 0.3492 | 0.3504 | 0.3492 |
|  | NAs | **0.2635** | **0.2581** | **0.2575** | **0.2582** |
| GINE | VNs | 0.3665 | 0.3614 | 0.3653 | 0.3687 |
|  | NAs | **0.2624** | **0.2565** | **0.2580** | **0.2598** |
| GCNII | VNs | 0.3686 | 0.3644 | 0.3648 | 0.3632 |
|  | NAs | **0.2670** | **0.2577** | **0.2551** | **0.2606** |
| GatedGCN | VNs | 0.3425 | 0.3398 | 0.3409 | 0.3374 |
|  | NAs | **0.2596** | **0.2553** | **0.2467** | **0.2473** |

As can be seen from Tab. 17 and Tab. 18, neural atoms achieve consistently and significantly better performance than the virtual nodes approach, regardless of their number. In both datasets, a larger number of neural atoms could lead to a better performance. Whereas virtual nodes achieve almost identical performance with increasing numbers, even with the pair-wise connections among them.

We speculate that such a phenomenon is because multiple virtual nodes might not learn representative subgraph patterns, which is crucial for the model to learn long-range interaction. The poor performance of multiple virtual nodes might be caused by the overwhelming aggregated information from all atoms within the graph, which leads to over-squashing and decreases the quality of the virtual node embeddings. This aligns with the description in the "Information aggregating" in Tab. 16. In addition, the virtual nodes are simply connected to all atoms within the molecular graph without considering their discrepancy. This could encourage the similarity among atom embeddings, which leads to poor performance. This aligns with the description in the "Backward projection" in Tab. 16.

Thus, we could claim that adopting and increasing the number of virtual nodes does not bring a noticeable improvement, whereas neural atoms could alleviate these issues and achieve better performance.

## F   LONG RANGE INTERACTION EXAMPLES

The long-range interaction affects the surface area of the molecule or contributes to the formation of hydrogen bonds, which in turn affects the properties, such as melting point, water affinity, viscosity, of the molecule (Ying et al., 2021; Liu et al., 2022a; Stärk et al., 2021; Gromiha & Selvaraj, 1999). Understanding and quantifying these long-range interactions is crucial in chemistry, physics, and materials science, as they influence the behavior and properties of molecules, materials, and biological systems. Here, we show different types of LRI and their properties.

- **Van der Waals Force** is a weak attractive interaction that occurs between all atoms and molecules. These forces arise due to temporary fluctuations in electron distribution, creating temporary dipoles. Van der Waals forces include London dispersion forces (arising from instantaneous dipoles), dipole-dipole interactions, and induced dipole-induced dipole interactions. These forces can act over relatively long distances and are responsible for the condensation of gases into liquids.

- **Hydrogen Bond** is a special type of dipole-dipole interaction that occurs when hydrogen is bonded to a highly electronegative atom (such as oxygen, nitrogen, or fluorine) and is attracted to another electronegative atom in a nearby molecule. Hydrogen bonds are relatively strong compared to other long-range interactions and play a crucial role in the structure and properties of water, DNA, and proteins.

- **Electrostatic Interaction**, also known as Coulombic interactions, occur between charged particles. While ionic bonds are a type of strong electrostatic interaction, long-range electrostatic interactions can also occur between charged ions or polar molecules that are not directly bonded to each other. These interactions can be both attractive and repulsive, depending on the charges involved.

- **Dispersion Interaction**, also known as London dispersion forces, is a component of van der Waals forces. They arise from temporary fluctuations in electron distribution and can act between all molecules, even non-polar ones. These forces can be relatively weak but can accumulate to have a significant impact on molecular interactions.

- **Magnetic Interaction** is a long-range magnetic interaction that can occur between magnetic moments associated with atoms or ions. These interactions are responsible for the behavior of ferromagnetic and antiferromagnetic materials.

## G    THE TRAINING CURVES

We provide the training curves for the methods in Tab. 2, shown as Fig. 7 to 15 with a smoothing ratio of 0.9.

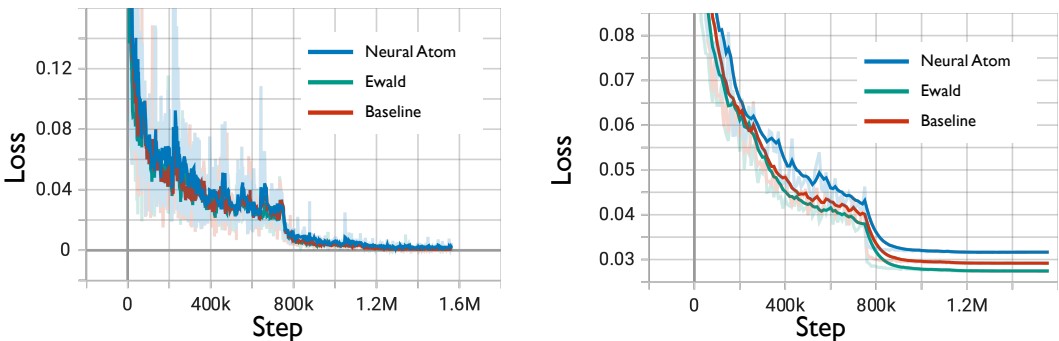

Figure 7: Training and validation loss curves visualizations for DimeNet++ (1) the training loss curve, (2) the validation loss curve.

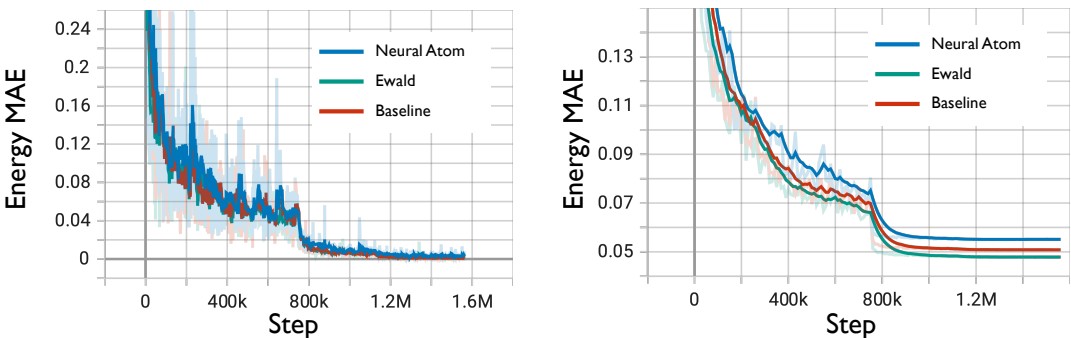

Figure 8: Training and validation energy MAE curves visualizations for DimeNet++ (1) the training MAE curve, (2) the validation MAE curve.

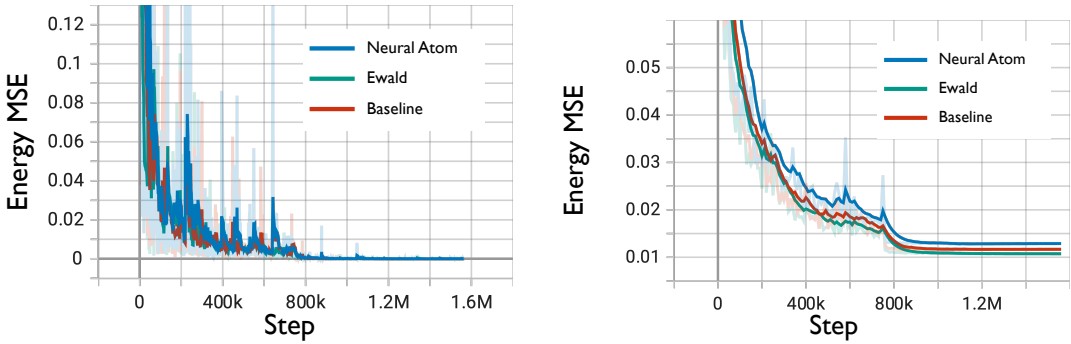

Figure 9: Training and validation energy MSE curves visualizations for DimeNet++ (1) the training MSE curve, (2) the validation MSE curve.

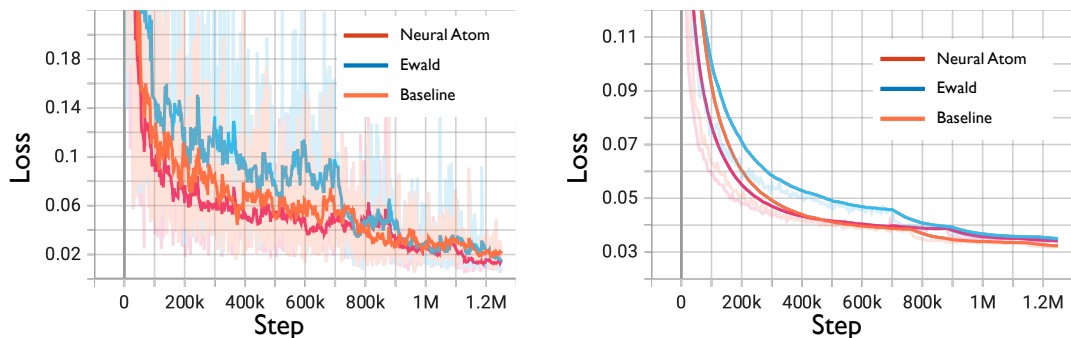

Figure 10: Training and validation loss curves visualizations for PaiNN (1) the training loss curve, (2) the validation loss curve.

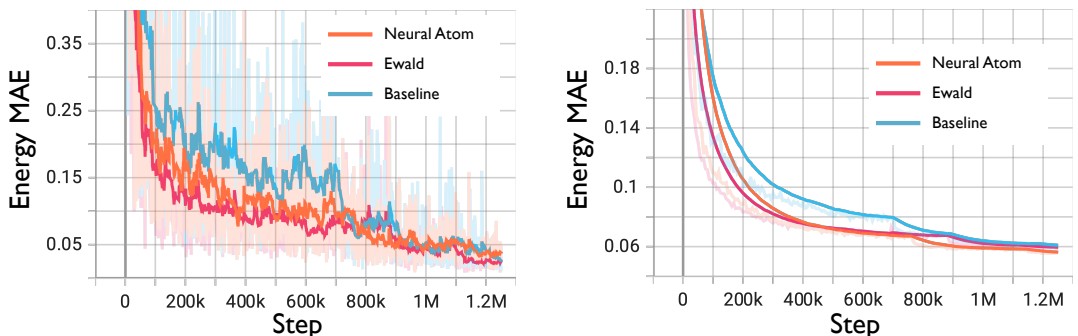

Figure 11: Training and validation energy MAE curves visualizations for PaiNN (1) the training MAE curve, (2) the validation MAE curve.

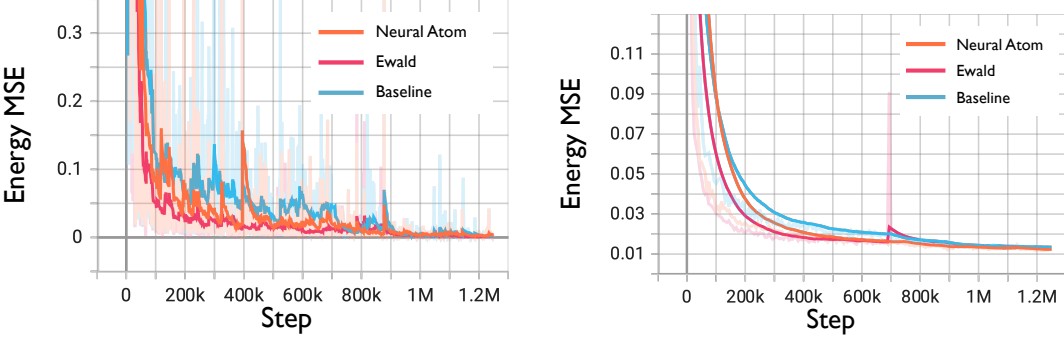

Figure 12: Training and validation energy MSE curves visualizations for PaiNN (1) the training MSE curve, (2) the validation MSE curve.

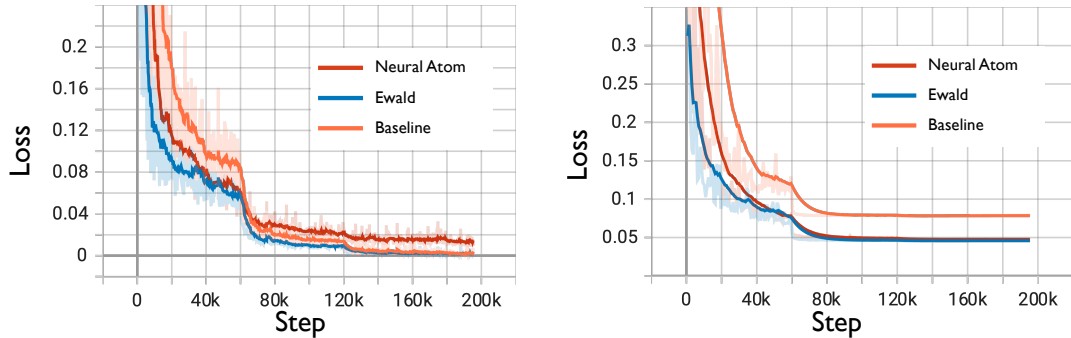

Figure 13: Training and validation loss curves visualizations for SchNet (1) the training loss curve, (2) the validation loss curve.

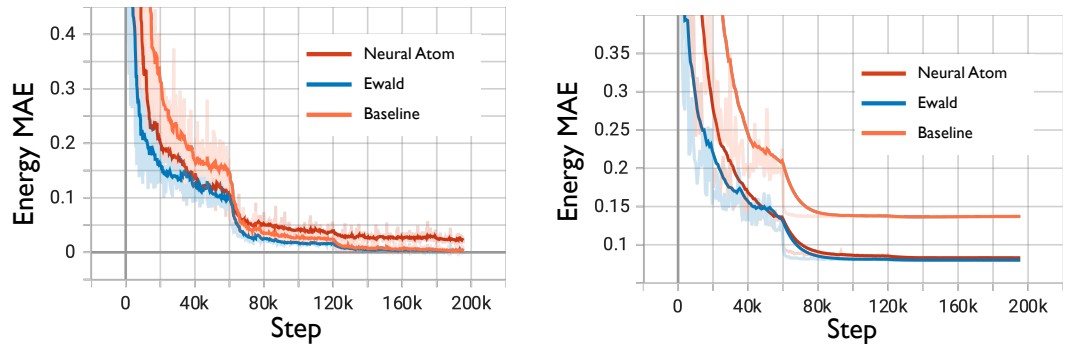

Figure 14: Training and validation energy MAE curves visualizations for SchNet (1) the training MAE curve, (2) the validation MAE curve.

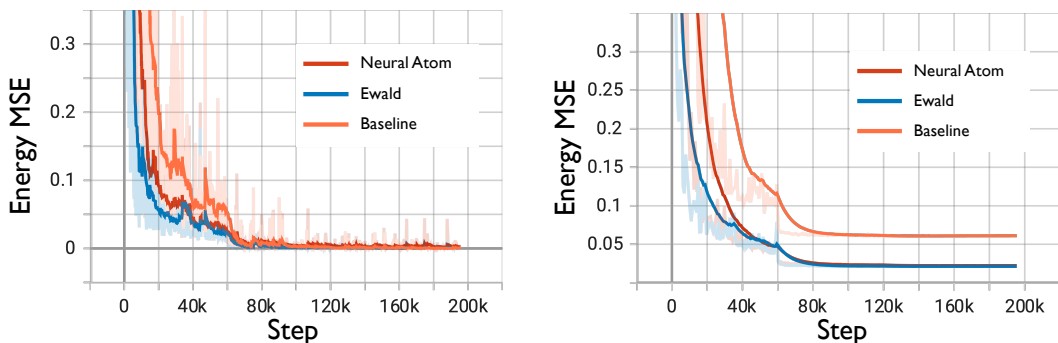

Figure 15: Training and validation energy MSE curves visualizations for SchNet (1) the training MSE curve, (2) the validation MSE curve.

## H  NEURAL ATOMS ASSIGNMENT AND INTERACTION VISUALIZATION

We visualize the interaction strength for the Mutagenicity dataset. Specifically, we extract the attention weight for both the allocation matrix $\hat{A}$ (upper figure), the attention weights for different neural atoms, which we denote as interaction matrix (lower left figure), and the original molecular graph with atom indices (lower right figure). We adopt a two-layer GIN with four neural atoms and extract the $\hat{A}$ from the last layer for visualization.

Shown as Fig. 16 to 30, the neural atoms aggregate information from different atoms within the molecular graph with varying attention strength. Such attention patterns allow the neural atoms to ensure the diversity of aggregated information by assigning different weights to different atom information. The interactions among neural atoms are established via the neural atom numbered one, which shows weak attention strength for all atoms in the original graph. Such an attention pattern indicates that the model learns to leverage the communication channel, i.e., the neural atom numbered one, to bridge the atoms with potential long-range interactions.

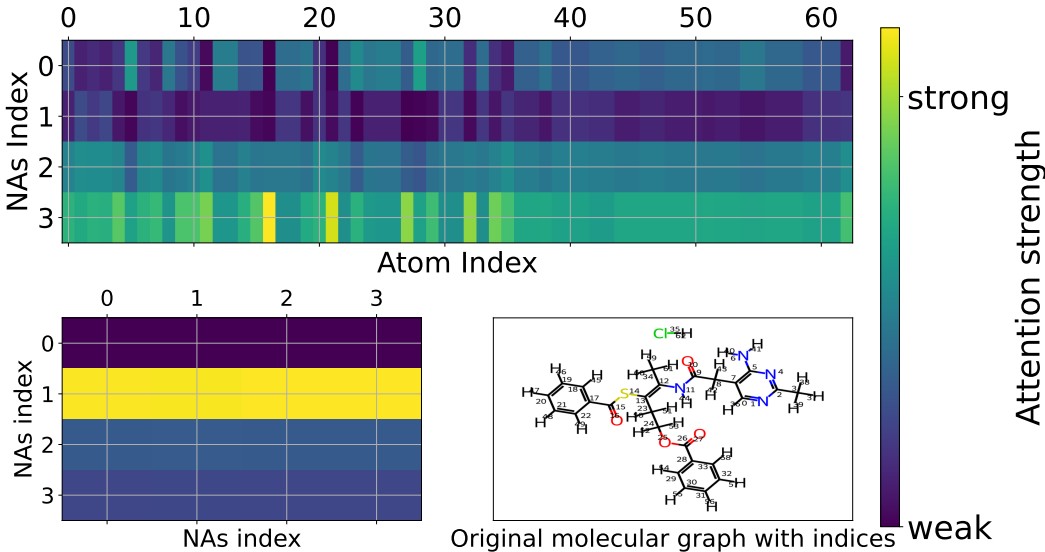

Figure 16: Mutagenicity test set index-7.

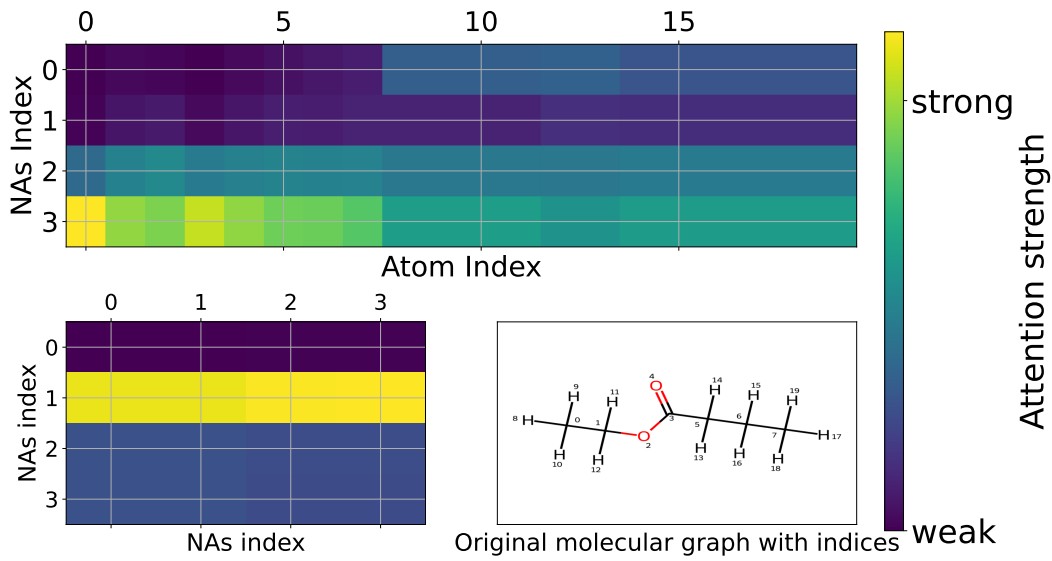

Figure 17: Mutagenicity test set index-17.

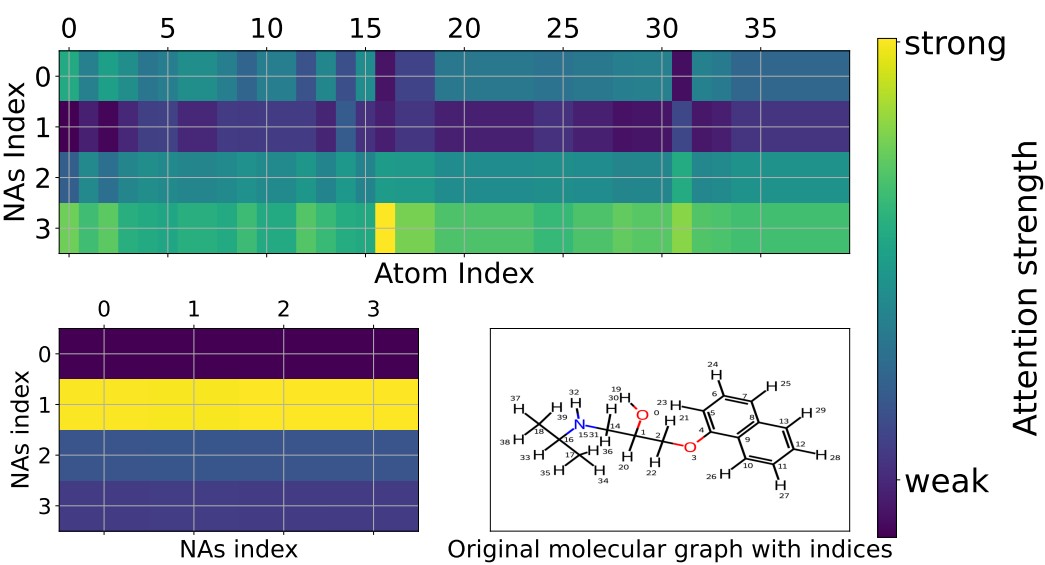

Figure 18: Mutagenicity test set index-18.

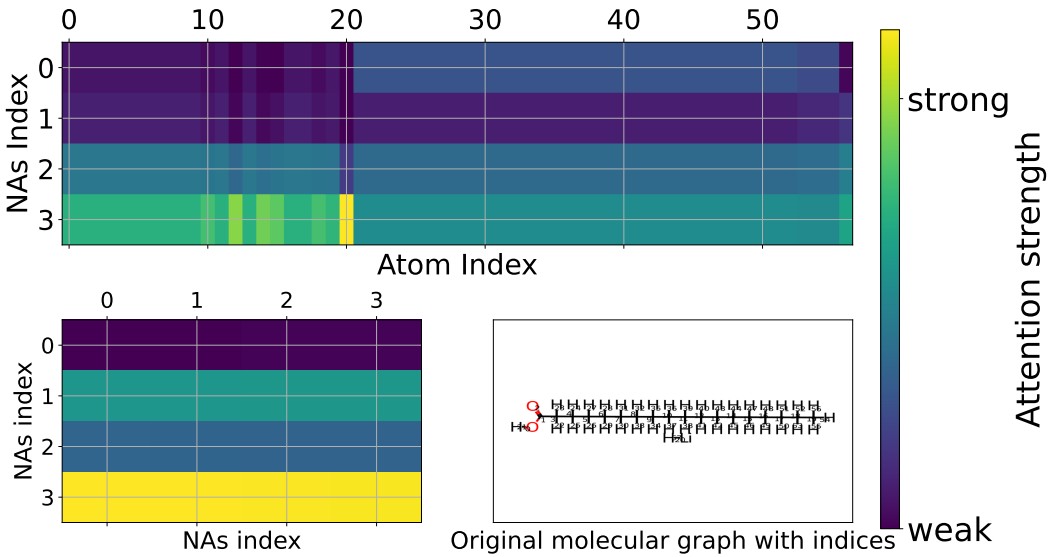

Figure 19: Mutagenicity test set index-24

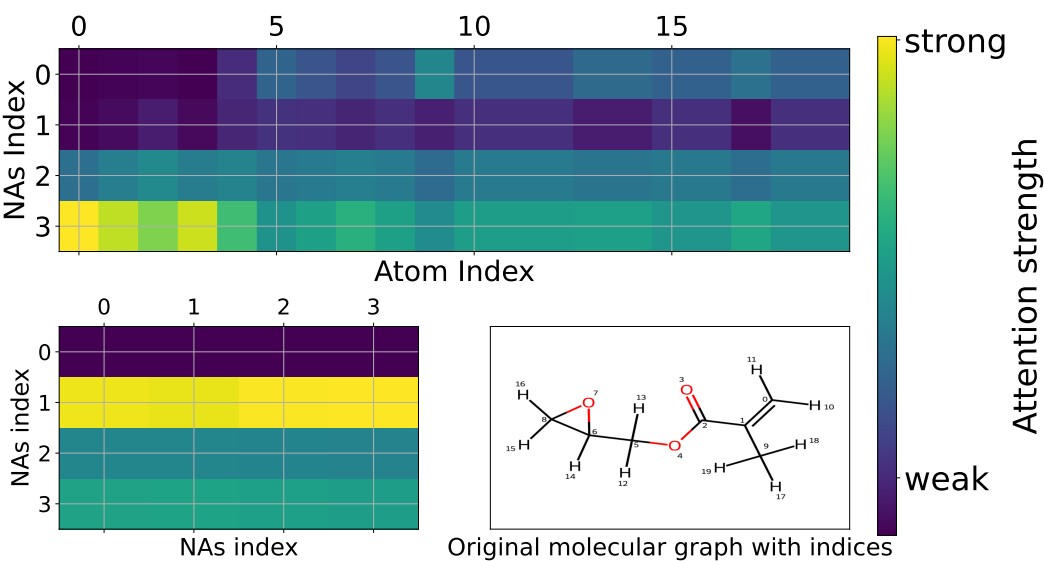

Figure 20: Mutagenicity test set index-26.

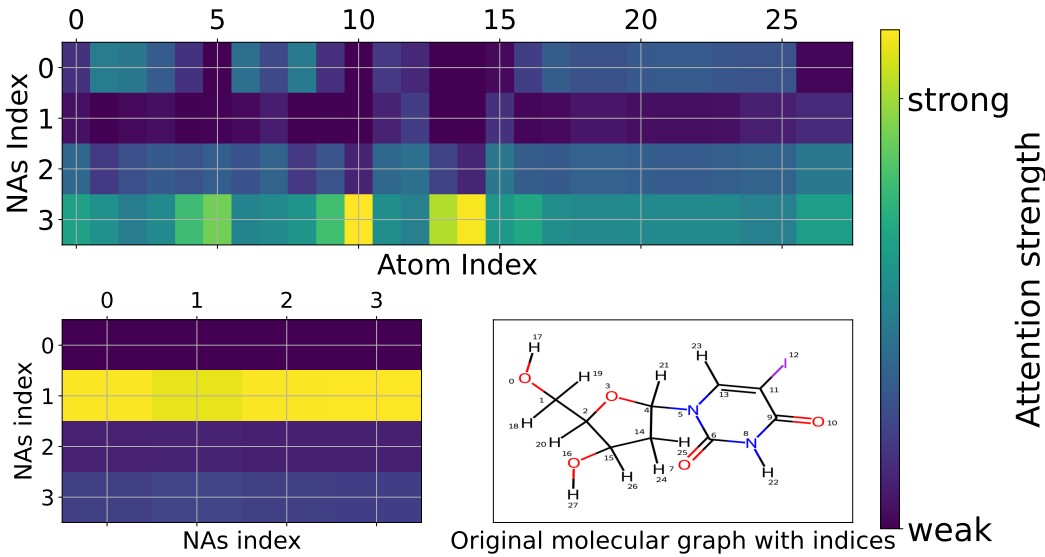

Figure 21: Mutagenicity test set index-28.

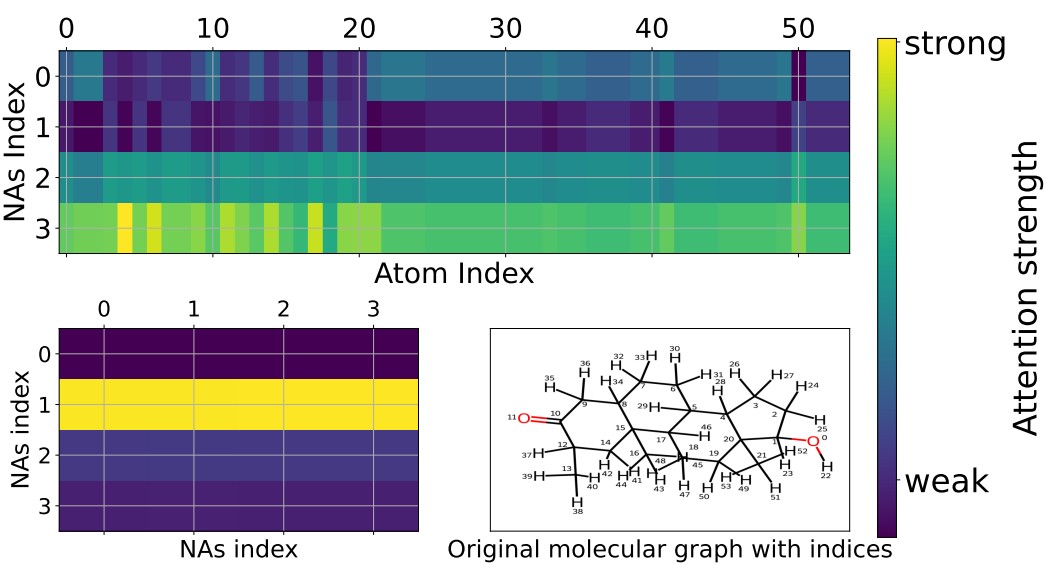

Figure 22: Mutagenicity test set index-29.

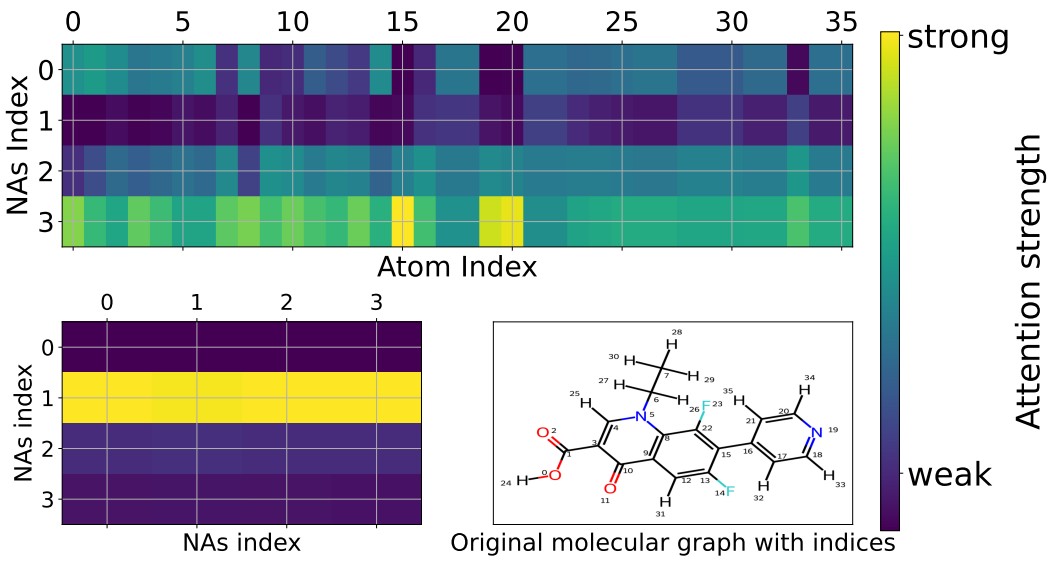

Figure 23: Mutagenicity test set index-32.

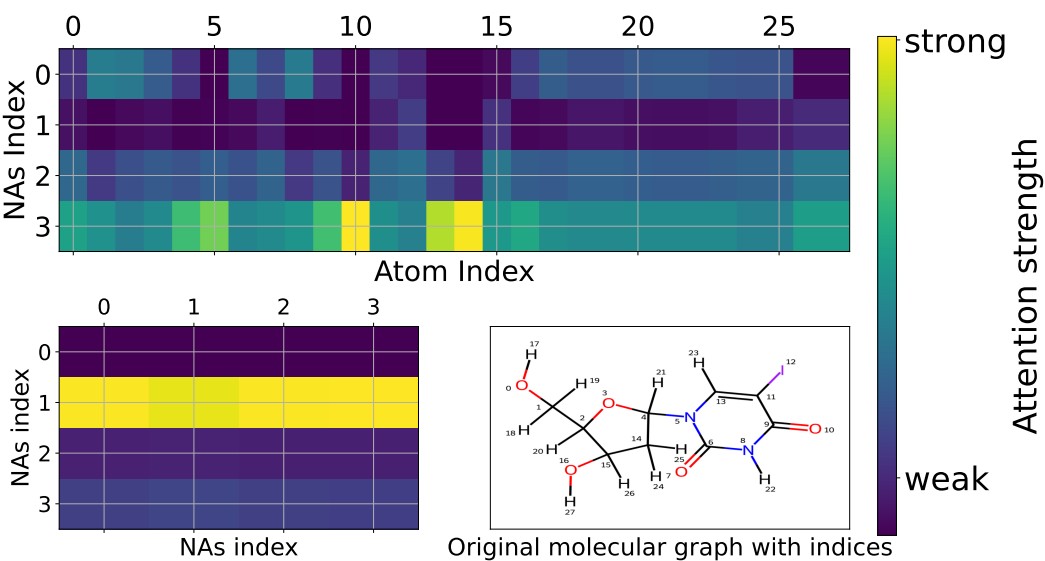

Figure 24: Mutagenicity test set index-36.

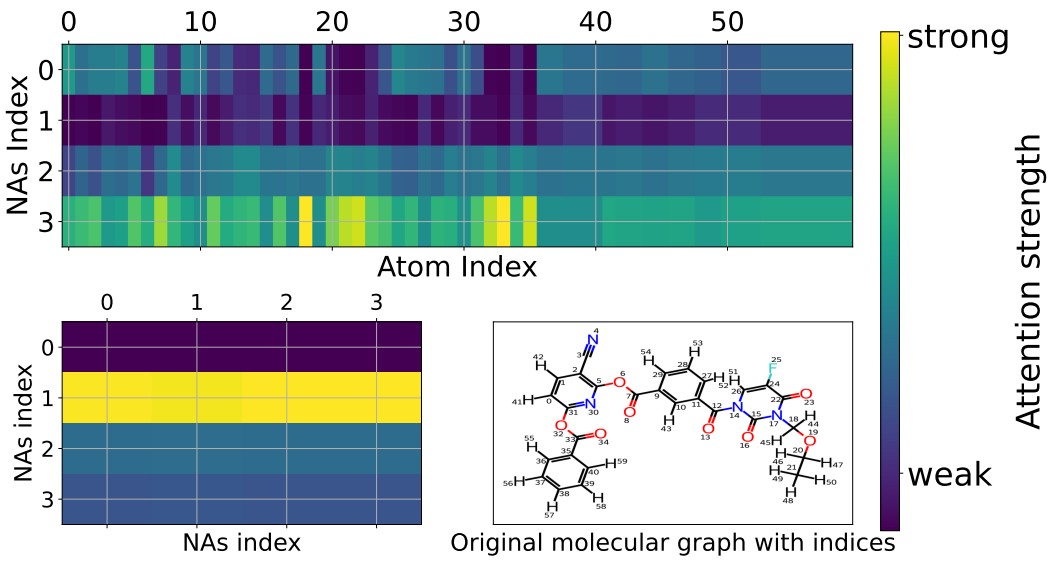

Figure 25: Mutagenicity test set index-64.

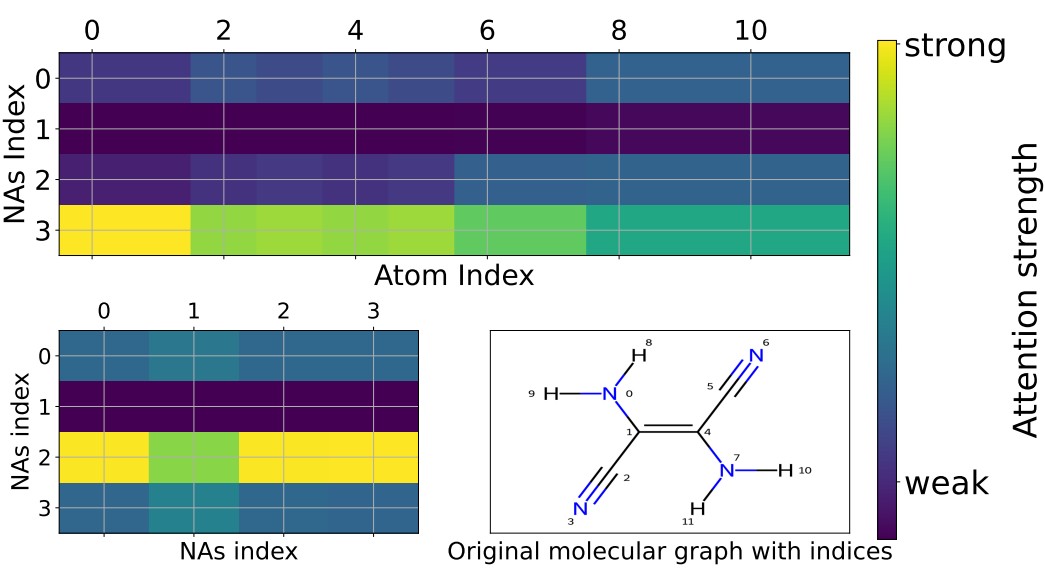

Figure 26: Mutagenicity test set index-77.

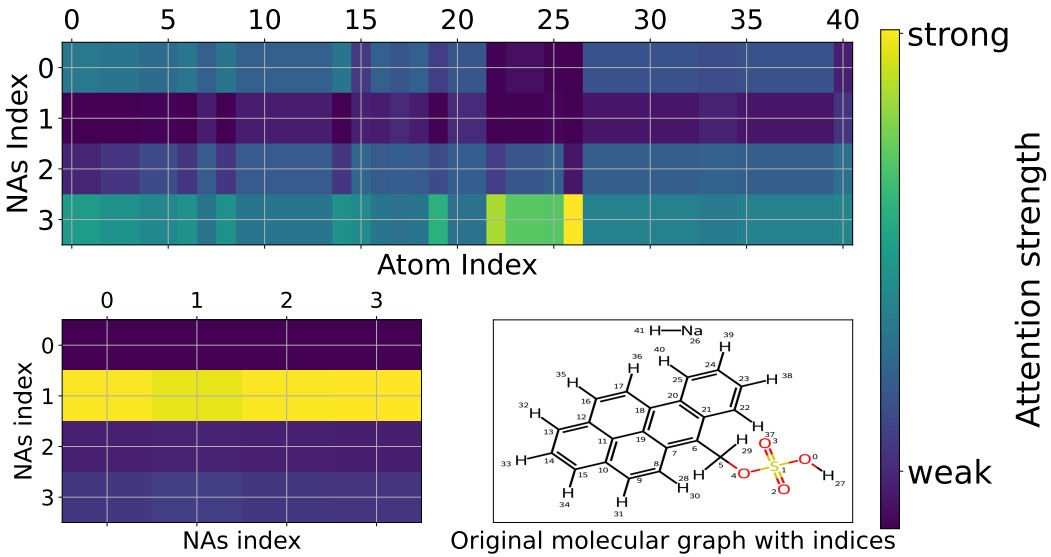

Figure 27: Mutagenicity test set index-88.

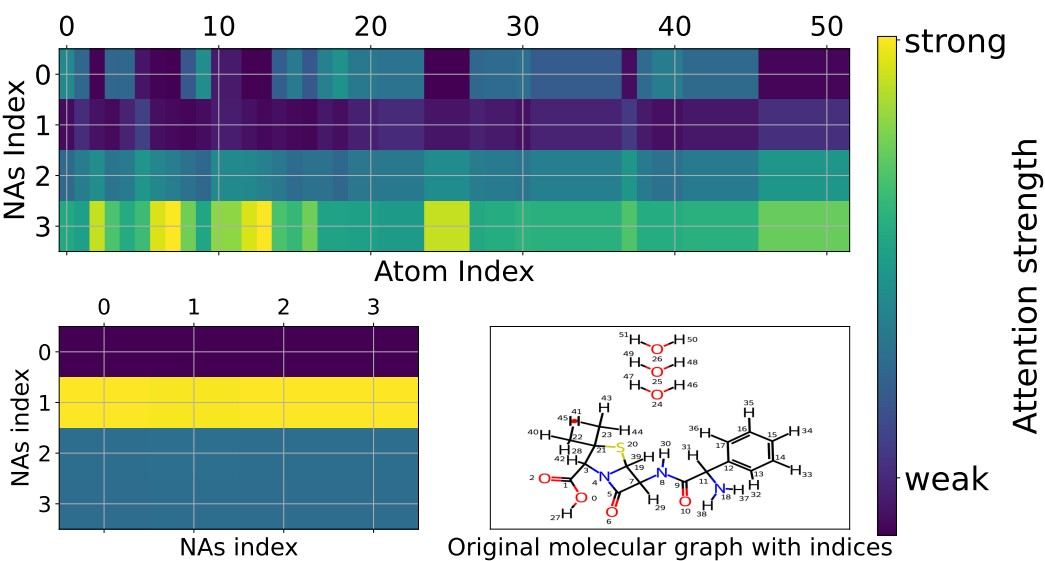

Figure 28: Mutagenicity test set index-99.

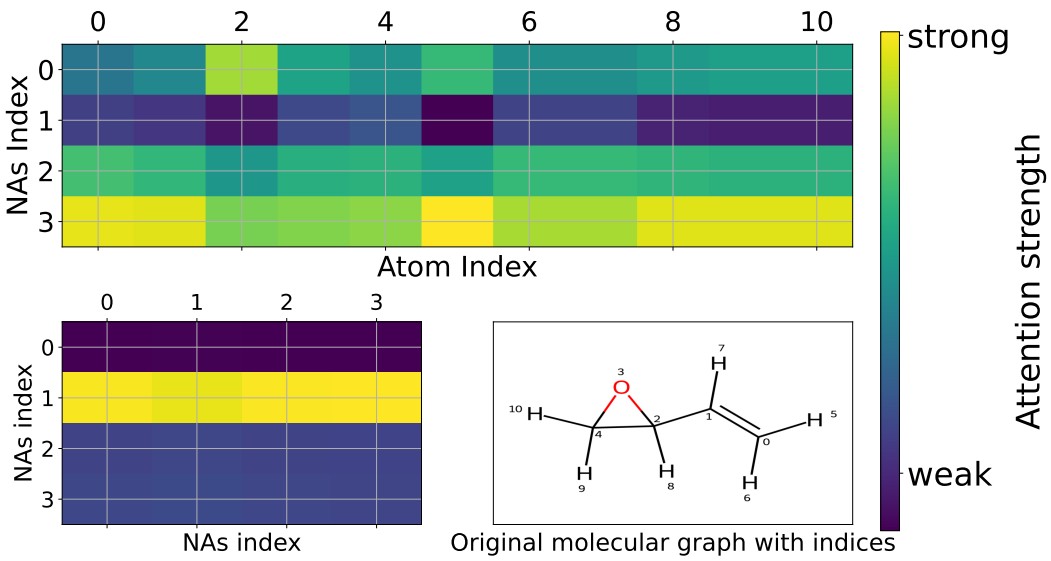

Figure 29: Mutagenicity test set index-211.

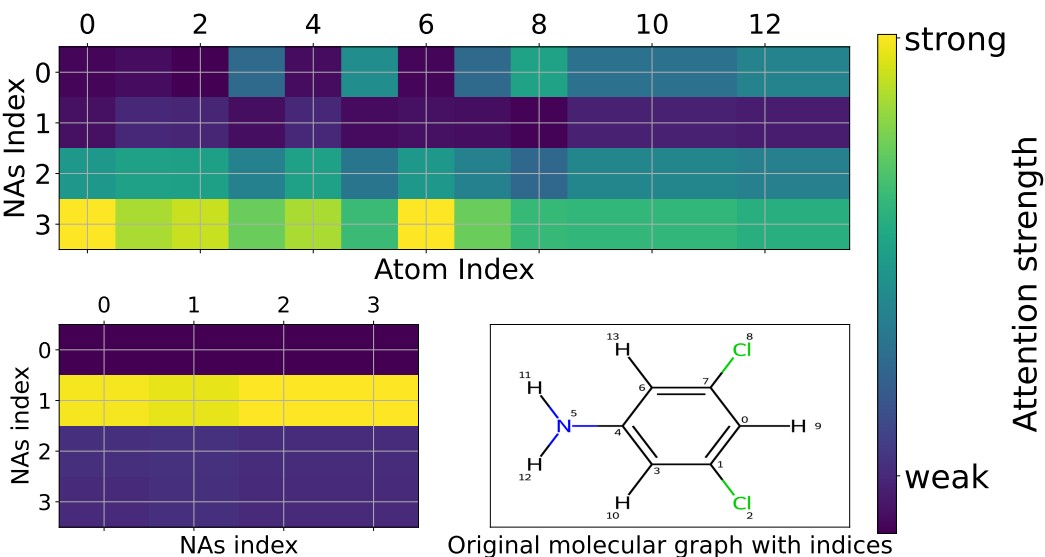

Figure 30: Mutagenicity test set index-399

