# OpenReview forum: "Neural Atoms: Propagating Long-range Interaction in Molecular Graphs through Efficient Communication Channel"
_ICLR.cc/2024/Conference — ICLR 2024 poster_

### Official Review · Reviewer_2N34 · 2023-10-28

**Soundness:** 2 fair
**Presentation:** 2 fair
**Contribution:** 2 fair
**Rating:** 6
**Confidence:** 4

**Summary:**

Summary:
This paper proposes a method called "Neural Atoms" to help graph neural networks (GNNs) better capture long-range interactions in molecular graphs. The key ideas can be understood as: (i) introduce a small set of "neural atoms" that group together subsets of the original atoms in the molecule. This reduces long interaction paths to a single hop between neural atoms. (ii) use attention mechanisms to learn how to group atoms into neural atoms; and to exchange information between the neural atoms. (iii) enhance the atom representations by mixing in information from the neural atoms, allowing GNNs to capture long-range interactions.

Overall, the paper seems to formalize the concept of neural atoms as virtual atoms that represent clusters of real atoms. In concept, this resembles similar to DiffPool (Ying et al., 2018 reference in the paper). however their implementation and purpose can be differently understood.

**Strengths:**

Strengths:
(a) simple and architecture-agnostic method that can enhance any GNN.
(b) reduces long-range interactions to single hop, avoiding path explosion issue.
(c) outperforms baselines on molecular graph tasks needing long-range modeling.
(d) visualizations show neural atoms learn meaningful groupings aligned with interactions.

**Weaknesses:**

Weaknesses:
(a) does not utilize 3D coordinate information available for some molecules.
(b) mainly considers intra-molecular, not inter-molecular interactions.
(c) hyperparameter tuning needed for number of neural atoms, as the problem with pooling operation known earlier in graph learning literature.
(d) lacks strong theory on optimal atom groupings.

**Questions:**

Questions:
(a) How are the neural atom groupings initialized? Are they randomly assigned?
(b) is there a theoretical justification for why this approach models long-range interactions well?
(c) could 3D coordinate data be incorporated to better model distance-dependent interactions?
(d) how does this approach scale to very large molecular graphs?

---

> ### Author Response · Authors · 2023-11-20
> **Response to Reviewer 2N34 (1/2)**
>
> We thank the reviewer 2N34 for the valuable and positive feedback. We addressed all the comments. Please find the detailed responses below. Any further comments and discussions are welcomed!
>
> **Q1:** Mainly focus on the 2D inter-molecular interaction. There is less emphasis on the inter-molecular interactions and not utilizing 3D coordinate information.
>
> > does not utilize 3D coordinate information available for some molecules; mainly considers intra-molecular, not inter-molecular interactions.
>
> **Reply:** Thanks for the valuable review! We would like to clarify the reason for not adopting 3D coordinate information and the consideration of inter-molecular interactions. Due to the time limitation, we are unable to incorporate the 3D information into the neural Atom mechanism, **as the 3D coordinate position of the neural atom needs to be carefully initialized and handled to avoid potential bias from the 3D position.** Here, we directly adopt the neural atoms to the 3D scenario without 3D coordinate information and provide the comparison with Ewald-based GNN [1] (denoted as Ewald Block) on the OE62 [2] dataset. We follow the same setting from [1] and use SchNet [3] as the backbone GNN. Results show that **our method achieves competitive performance even with the absence of 3D coordinate information**, showing the generalization and effectiveness of neural atoms.
>
> |                   | Energy MAE $\downarrow$ | Energy MSE $\downarrow$ | Using 3D information |
> | :---------------: | :---------------------: | :---------------------: | :------------------: |
> | Baseline (SchNet) |         0.1351          |         0.0658          |     $\checkmark$     |
> |   + Ewald Block   |       **0.0811**        |       **0.0301**        |     $\checkmark$     |
> |   + Neural Atom   |       **0.0834**        |       **0.0309**        |       $\times$       |
>
> We have added this discussion to our draft in Appendix D.
>
> [1] Ewald-based Long-Range Message Passing for Molecular Graphs. In ICML, 2023
>
> [2] Atomic structures and orbital energies of 61,489 crystal-forming organic molecules. In Scientific Data, 2020.
>
> [3] SchNet: A continuous-filter convolutional neural network for modeling quantum interactions. In NeurIPS, 2017.
>
> **Q2:** The requirement of the number of neural atoms for hyperparameter tuning.
>
> > hyperparameter tuning needed for number of neural atoms, as the problem with pooling operation known earlier in graph learning literature.
>
> **Reply:** Thanks for the valuable review. We would like to clarify further the setting of the proposed neural atoms. We do not directly finetune the number of neural atoms but define the number as the proportion of the average number of nodes for the specific dataset. For example, we use only 22 neural atoms for GCN in both Peptides-func and Peptides-struct datasets, which is far less than the original number of atoms (around 150 per graph). We update our hyperparameters with the specific number of neural Atoms used. We kindly refer the reviewer to Table 4 and Table 7 to Table 9 for more detailed hyperparameter settings, which show that our method can exceed the baseline with even one-tenth number of nodes w.r.t. the average number of nodes in the original graph.

---

> ### Author Response · Authors · 2023-11-20
> **Response to Reviewer 2N34 (2/2)**
>
> **Q3:** A lack of a fully developed theoretical framework for determining optimal atom groupings.
>
> > lacks strong theory on optimal atom groupings.;
> >
> > is there a theoretical justification for why this approach models long-range interactions well?
>
> **Reply:**  Thank you for the question. We have to admit the (1) unavailability of the theory of the atom groupings strategy and (2) potential useful discussion and future works.
>
> (1) Since we mainly focus on the methodology and the experimental evaluation of our method, we have not derived a theoretical justification for the proposed method. However, we did explore most of the research about the long-range interaction and grouping theory on the molecular graphs, but the available information is limited. As such, **to the best of our knowledge, there might not be a strong theory about the atom grouping strategy since the functional groups of different molecular graphs vary [1].** We leave this theoretical justification for future works.
> (2) **Alternatively, there is some research on the impact of the virtual node.** For example, [2] studies the benefit of introducing single or multiple virtual nodes for link prediction tasks and provides a theoretical claim from the graph random walk perspective; [3] investigates the relationship between GNN and Graph Transformers via the bridge of the virtual node, and provide the assumptions and theory that guarantee the approximation of GNN with virtual node to the Graph Transformer. Our neural atoms might establish a similar theoretical framework like [2], as we reduce the communication distance that could affect the result of random walk.
>
> [1] Chemistry: the central science. Brown, Theodore (2002). ISBN 0130669970.
>
> [2] An analysis of virtual nodes in graph neural networks for link prediction. In LoG, 2022.
>
> [3] On the Connection Between MPNN and Graph Transformer, In ICML, 2023.
>
> **Q4:** The initialization of the assignment of neural atoms.
>
> > How are the neural atom groupings initialized? Are they randomly assigned?
>
> **Reply:** Thanks for your question. The allocation matrix is computed as ${\hat{A}} = \sigma(Q^{(\ell)}_{\text{NA}} K^\top) \in \mathbb{R}^{K \times N}$. Therefore, the $\hat{{A}}$ is dynamically calculated according to the embedding similarity between the neural atoms and the atom within the molecular graph.
>
>
> **Q5:** Cooperating with 3D coordination information.
>
> > could 3D coordinate data be incorporated to better model distance-dependent interactions?
>
> **Reply:** It is possible to incorporate the 3D coordinate information if there is an available data source. One can employ a 3D Transformer like Point Transformer to process the 3D molecular graph, and the neural atoms can also be equipped with 3D coordinate information to handle the position information. However, as this paper mainly focuses on 2D and intra-molecular graphs, 3D information is usually not provided on most of the benchmark datasets. In addition, due to time limitations, we are unable to incorporate the 3D information into the neural atom mechanism. The 3D coordinate position of the neural atom needs to be carefully initialized and handled to avoid potential bias from the 3D position.
>
> **Q6:** Stategy to scale the method to very large molecular graph.
>
> > how does this approach scale to very large molecular graphs?
>
> **Reply:** Thanks for the valuable question! To scale the neural atoms to the very large molecular graph, one could employ a linear Transformer like Performer [1] or BigBrid [2]. In addition, one could prune the connection between neural atoms and the atoms within the original molecular graph based on the attention score to reduce the potential computation burden. **We have highlighted this point in the revised submission for the potential extension to the broad applications.**
>
> The scale of the dataset we adopted, i.e., the Long-Range Graph Benchmark, is considered to be **larger than the general molecular graph dataset w.r.t. the graph size.** We show the comparison of the dataset statistic in the following table:
>
> |                | #Graphs | #Nodes per graph | #Edges per graph |            Task Type             | Metric  |
> | :------------: | :-----: | :--------------: | :--------------: | :------------------------------: | :-----: |
> |  ogbg-molhiv   | 41,127  |       25.5       |       27.5       |   Graph binary classification    | ROC-AUC |
> |  ogbg-molpcba  | 437,929 |       26.0       |       28.1       |   Graph binary classification    |   AP    |
> |  LRGB-pepfunc  | 15,535  |      150.94      |      307.3       | Graph multi-label classification |   AP    |
> | LRGB-pepstruct | 15,535  |      150.94      |      307.3       |   Graph multi-label regression   |   MAE   |
>
> [1] Rethinking attention with performers. In ICLR, 2021.
>
> [2] Big Bird: Transformers for Longer Sequences. In NIPS, 2021.

---

> ### Author Response · Authors · 2023-11-22
> **Would you mind checking our responses and confirming whether you have any further questions?**
>
> Dear Reviewer 2N34,
>
> Thanks very much for your time and valuable comments.
>
> In the rebuttal period, we have provided detailed responses to all your comments and questions point-by-point for the unclear presentations. Specifically, we provided detailed explanations to further clarify
>
> - Mainly focus on the 2D inter-molecular interaction. There is less emphasis on the inter-molecular interactions and not utilizing 3D coordinate information. (Q1)
> - The requirement of the number of neural atoms for hyperparameter tuning. (Q2)
> - A lack of a fully developed theoretical framework for determining optimal atom groupings. (Q3)
> - The initialization of the assignment of neural atoms. (Q4)
> - Cooperating with 3D coordination information. (Q5)
> - Stategy to scale the method to very large molecular graph. (Q6)
>
> Would you mind checking our responses and confirming whether you have any further questions?
>
> Any comments and discussions are welcome!
>
> Thanks for your attention and best regards.

---

> > ### Comment · Reviewer_2N34 · 2023-11-22
> > **Response to rebuttal**
> >
> > Dear authors, thank you very much for your detailed responses to my questions and clarifying issues on the aspects of grouping, use of 3D information, scalability, and pointers to the relevant literature on virtual nodes, among others. I acknowledge that I read the responses, along with the revised manuscript. I will retain my score based on the papers' contributions, while inclining for acceptance.

---

### Official Review · Reviewer_9FRM · 2023-10-31

**Soundness:** 3 good
**Presentation:** 3 good
**Contribution:** 3 good
**Rating:** 8
**Confidence:** 3

**Summary:**

This paper proposes a technique called Neural Atom that tries to abstract a cluster of nodes into a single node and subsequently leverage these condensed nodes for exchanging information that may not be achievable within the original molecule graphs. The authors validate the effectiveness of these proposed techniques through comprehensive experimentation conducted on three distinct datasets and employing various GNNs.

**Strengths:**

- The authors have made commendable efforts to clarify how their proposed algorithm works by providing theoretical explanations, which is commendable.
- They have also used real-world case studies to show how their method functions in practice.

**Weaknesses:**

- The incorporation of virtual atoms is not a novel concept, as it has previously been applied graph research as early as 2017 [r1, r2, r3]. It would be beneficial for the authors to engage in a discussion regarding how their proposed techniques differ from these referenced works. Furthermore, it is worth noting that there is a concurrent study that outlines a similar pipeline [r4]. Given the potential significance of this similarity, it would be valuable to include a discussion of this related work.

- The proposed approach has the potential to be employed across various types of graphs, as the concept of "virtual atoms" could be (and already have been) utilized  in other datasets like the OGB benchmarks. However, the authors did not explore this possibility in their work.
- The quality of writing in the paper seems to get worse as you read further, especially in Section 4. There are a lot of grammar mistakes in Section 4.2. Additionally, there are some terms like "ratio" in Table 4 and "varying proportion" in the last paragraph on page 7 that are introduced without prior explanation. This gives the impression that the paper was completed hastily.

[r1] Molformer: Motif-Based Transformer on 3D Heterogeneous Molecular Graphs, AAAI 2021

[r2] Neural Message Passing for Quantum Chemistry, ICML 2017

[r3] An analysis of virtual nodes in graph neural networks for link
prediction, LoG 2022

[r4] On the Connection Between MPNN and Graph Transformer

**Questions:**

I have a couple of questions about Figure 5. Can atoms within the same neural atoms share information? If so, I'm wondering whether information exchange between atoms from different neural atoms is actually more than between atoms within the same neural atoms.

I am open to adjust my scores after the discussion with the authors.

---

> ### Author Response · Authors · 2023-11-20
> **Response to Reviewer 9FRM (1/3)**
>
> We thank reviewer 9FRM for the valuable and positive feedback. We addressed all the comments. Please kindly find the detailed responses below. Any further comments and discussions are welcomed!
>
> **Q1:** Further discussion of related papers [r1-r4].
>
> > The incorporation of virtual atoms is not a novel concept, as it has previously been applied to graph research as early as 2017 [r1, r2, r3]. It would be beneficial for the authors to engage in a discussion regarding how their proposed techniques differ from these referenced works.  Furthermore, it is worth noting that there is a concurrent study that outlines a similar pipeline [r4]. Given the potential significance of this similarity, it would be valuable to include a discussion of this related work.
>
> **Reply:** Thanks for the valuable question and the referenced paper. We would like to clarify further the difference between the neural atom and the method used in these papers. We have carefully revised the draft and added the recommended references to the draft following your suggestions in Appendix F.
>
> [r1] Molformer combines molecular motifs and 3D geometry information by heterogeneous self-attention to create expressive molecular representations. The paper uses a virtual atom as a starting point to extract the graph representation for downstream graph-level tasks. The paper proposes attentive farthest point sampling for sampling atoms in 3D space not only according to their coordinates but also their attention score. **However, the virtual atom they utilize does not participate in the message aggregation nor graph-level representation extraction, as they claim to "locate a virtual node in 3D space and build connections to existing vertices."** As such, the potential long-range interaction they capture might be due to the atom pair-wise heterogeneous self-attention, which differs from our method, where the long-range interaction is captured by both the attention mechanism (step.1 in Figure 3) and the interaction among the neural atoms (step.2 in Figure 3).
>
> [r2] first introduces the concept of Message Passing Neural Networks (MPNN) to develop a unified framework for predicting molecular properties. The paper introduces the "virtual node" as an argument for global information extraction. The virtual node, connected to all other nodes within the graph, acts as the global communication channel, enhancing the model's ability to capture long-range interactions and dependencies in molecular graphs. The authors experimented with a "master node" connected to all other nodes, serving as a global feature aggregator. This approach showed promise in improving the model's performance, especially in scenarios where spatial information, e.g., 3D coordination, is limited or absent.
>
> [r3] study the benefit of introducing single or multiple virtual nodes for link prediction tasks. Virtual node, traditionally thought to serve as aggregated representations of the entire graph, is connected to subsets of graph nodes based on either randomness or clustering mechanism. Such methodology significantly increases the expressiveness and reduces under-reaching issues in MPNN. The study reveals that virtual nodes, when strategically integrated, can provide stable performance improvements across various MPNN architectures and are particularly beneficial in dense graph scenarios. Their virtual node differs from the neural atom regarding the grouping strategy and the information-exchanging mechanism.
>
> [r4] investigates the relationship between MPNN and Graph Transformers (GT) via the bridge of the virtual node. It demonstrates that MPNN augmented with virtual nodes can approximate the self-attention layer of GT. Under certain circumstances, the paper provides a construction for MPNN + VN with O(1) width and O(n) depth to approximate the self-attention layer in GTs. The paper provides valuable insight into understanding the theoretical capabilities of MPNN with virtual nodes in approximating GT. Compared to neural atoms, we do not focus on establishing a theoretical connection between MPNN and GT. Instead, we are interested in leveraging the attention mechanism's ability to construct an interaction subspace constructed by the neural atoms. As such, the subspace acts as a communication channel to reduce interaction distances between nodes to a single hop.
>
> [r1] Molformer: Motif-Based Transformer on 3D Heterogeneous Molecular Graphs. In AAAI, 2021.
>
> [r2] Neural Message Passing for Quantum Chemistry. In ICML, 2017.
>
> [r3] An analysis of virtual nodes in graph neural networks for link prediction. In LoG, 2022.
>
> [r4] On the Connection Between MPNN and Graph Transformer, In ICML, 2023.

---

> ### Author Response · Authors · 2023-11-20
> **Response to Reviewer 9FRM (2/3)**
>
> **Q2:** Discussion of "virtual node" in OGB benchmarks.
>
> > The proposed approach has the potential to be employed across various types of graphs, as the concept of "virtual atoms" could be (and already have been) utilized in other datasets like the OGB benchmarks. However, the authors did not explore this possibility in their work.
>
> **Reply:** Thanks for your valuable feedback. Here, we would like to clarify the difference between Nerual Atoms and other supernode methods. We will elaborate our justification from five: the **number of supernode or neural atoms, the information aggregating strategy, the interaction among them, the method to project the information back to the original graph**
>
> Both the pipeline of supernode and neural atoms for obtaining global graph information can be described as two steps.
>
> - **Information aggregating**: The information of atoms within the molecular graph is aggregated into either supernode or multiple neural atoms with pair-wise connection.
> - **Interaction among node/atoms**: The second step shows differences in interaction among node/atoms. Supernode commonly exists alone, which means there is **only one** super node and thus lacks the ability to interact with others like neural atoms.
> - **Backward projection:** The final step shows differences in terms of the interaction among nodes/atoms. As supernode commonly exists alone, it thereby lacks the ability to interact with others like neural atoms.
>
> We provide a detailed comparison of supernodes and neural atoms in the following table.
>
>
> |                                          |                          Super Node                          |                         Neural Atoms                         |
> | :--------------------------------------: | :----------------------------------------------------------: | :----------------------------------------------------------: |
> |             #Atoms / #Nodes              | 1 (In most cases, there is only one single supernode for aggregating the global graph information, as increasing the number of which might not lead to performance improvement.) | $K$ (Our neural atoms can be defined as the proportion of the average number of atoms of the molecular graph dataset, which is **significantly smaller than the original number of atoms. The more neural atoms, the better the performance.**) |
> |         Information aggregating          | The global pooling method, e.g., global sum/mean/max pooling, **treats all the information from nodes the same**, thus lacking diversity among different nodes in the graph. | The multi-head Attention mechanism allows **aggregating information with different weights according to the similarity between specific neural atoms and the original atoms**, which allows diversity among different atoms. |
> | Interaction among supernode/neural atoms | None. (None. (Since supernode usually exists alone, it thus lacks the ability to interact with others.) | A fully connected graph with an attention mechanism to bridge the neural atoms for information exchange. This allows the information located in a different part of the graph, even with a large hop distance, to share information based on their embedding similarities. |
> |           Backward projection            | Direct element-wise adding. The information of the supernode is directly added to the representation of each node, which fuses the information from the single supernode, which might lead to the similarity among different atoms. | Weighted combination. The neural atoms can fuse the information within according to the similarity score between t neural atoms and the atoms in the original molecular graph. Such a mechanism allows the model to obtain diversity representation for further purposes. |
>
> We have added this to our Appendix F; we kindly refer the reviewer to check our latest version of the draft.

---

> ### Author Response · Authors · 2023-11-20
> **Response to Reviewer 9FRM (3/3)**
>
> **Q3:** Writting quality and missing prior explanation.
>
> > The quality of writing in the paper seems to get worse as you read further, especially in Section 4. There are a lot of grammar mistakes in Section 4.2. Additionally, there are some terms like "ratio" in Table 4 and "varying proportion" in the last paragraph on page 7 that are introduced without prior explanation. This gives the impression that the paper was completed hastily.
>
> **Reply:** Thanks for pointing out these problems. Following your suggestions, we have thoroughly revised the submission to improve the writing and presentation, which are highlighted in blue. We kindly refer the reviewer to assess the updated version. Besides, the concept of “proportion” is introduced at the beginning of Sec 4.2, which refers to the number of neural atoms, as it is set according to the average number of atoms in the dataset. In addition, we remove the term “ratio” to eliminate the potentially misleading information.
>
> **Q4:** Question about the information interaction within same neural atoms.
>
> > I have a couple of questions about Figure 5. Can atoms within the same neural atoms share information?  If so, I'm wondering whether information exchange between atoms from different neural atoms is actually more than between atoms within the same neural atoms.
>
> **Reply:** To answer this question, we would first (1) clarify further the details of information exchange within and outside the neural atom. In addition, we provide (2) detailed elaboration on whether the information exchanged within and outside the neural atoms.
>
> **Yes, the atoms within the same neural atoms could share information.** As we aggregate the information via multi-head attention, and project back to the original atoms, the information from different atoms could share with each others within the same neural atoms through this process. However, it is difficult to quantify or visually explain the amount of information exchanged within a specific neural atom, as the exchange happens in the hidden embedding space.
>
> Alternatively, We visualize the interaction strength for the Mutagenicity dataset. Specifically, we extract the attention weight for both the allocation matrix $\hat{A}$ (upper figure), the attention weights for different neural atoms, which we denote as interaction matrix (lower left figure), and the original molecular graph with atom indices (lower right figure) in Appendix G of the updated draft.
>
>
> The neural atoms aggregate information from different atoms within the molecular graph with varying attention strength. Such attention patterns allow the neural atoms to ensure the diversity of aggregated information by assigning different weights to different atom information. The interactions among neural atoms are established via the neural atom numbered one, which shows weak attention strength for all atoms in the original graph. Such an attention pattern indicates that the model learns to leverage the communication channel, i.e., the neural atom numbered one, to bridge the atoms with potential long-range interactions.

---

> ### Author Response · Authors · 2023-11-22
> **Would you mind checking our responses and confirming whether you have any further questions?**
>
> Dear Reviewer 9FRM,
>
> Thanks very much for your time and valuable comments.
>
> In the rebuttal period, we have provided detailed responses to all your comments and questions point-by-point for the unclear presentations. Specifically, we provided detailed explanations to further clarify
>
> - Further discussion of related papers. (Q1)
> - Discussion of “virtual node” in OGB benchmarks. (Q2)
> - Writting quality and missing prior explanation. (Q3)
> - Question about the information interaction within the same neural atoms. (Q4)
>
> Would you mind checking our responses and confirming whether you have any further questions?
>
> Any comments and discussions are welcome!
>
> Thanks for your attention and best regards.

---

> ### Comment · Reviewer_9FRM · 2023-11-23
> **Acknowledgement**
>
> Thanks for providing the responses! They have addressed my concerns (mostly about clarifying the difference between this work and those previous).

---

### Official Review · Reviewer_4gc7 · 2023-11-01

**Soundness:** 2 fair
**Presentation:** 2 fair
**Contribution:** 1 poor
**Rating:** 6
**Confidence:** 3

**Summary:**

In this paper, the authors address a crucial challenge in drug discovery using Graph Neural Networks (GNNs) - the difficulty in capturing both short-range interactions (SRI) and long-range interactions (LRI) within molecular graphs. While current GNNs excel at modeling SRI, they struggle with LRI, essential for determining molecular properties. To overcome this limitation, the authors propose a novel approach. They introduce "Neural Atoms," abstract representations that amalgamate information from atomic groups within a molecule. By exchanging information among these neural atoms and projecting them back to atoms’ representations, they establish effective communication channels among distant nodes, reducing the interaction scope of node pairs to a single hop. The method's efficacy is validated through extensive experiments on three long-range graph benchmarks, demonstrating its ability to enhance any GNN in capturing LRI, a crucial step forward in molecular graph analysis for drug discovery. Additionally, the paper provides a physical perspective, establishing a connection between this method and the traditional LRI calculation method, Ewald Summation.

====================

During rebuttal, the authors provided more experimental results and more detailed discussions on several aspects. Therefore, I increase my rating.

**Strengths:**

- The problem of capturing long-range interactions in molecular graphs is interesting and important.

- The paper is generally well-written and almost clear everywhere.

- Experiments conducted on several datasets, to some extent, show the effectiveness of the proposed method in both graph-level and link-level tasks.

**Weaknesses:**

- The novelty of the proposed method is limited. For example, from the perspective of general graph machine learning, supernodes have been widely used which are the same as the concept of neural atoms.

- Some claims are controversial: in the Introduction, the authors claimed the disadvantages of transformers, especially the self-attention mechanism, but the way to project atom representations to neural atom representations in the proposed method still uses multi-head attention. Similarly, the controversy also happens in the running time experiment.

- There are some limitations in the experimental studies including:

1.  Experiments have been conducted on only three relatively small-size benchmark datasets. In fact, there are some larger and more recent benchmark datasets such as OC20 and OE62.
2.  Representative SOTA methods have not been compared, for instance [1].

[1] Ewald-based Long-Range Message Passing for Molecular Graphs, ICML 2023

**Questions:**

- From the comparison of the running time, there are no clear advantages of the proposed method compared to the transformers which conflicts with the claim in the Introduction (GTs are more computationally expensive than GNNs). What are the reasons? The multi-head attention used in step 1?

- What will be the performance on larger datasets? Can the proposed method beat more recent SOTA such as [1]?

[1] Ewald-based Long-Range Message Passing for Molecular Graphs, ICML 2023

---

> ### Author Response · Authors · 2023-11-20
> **Response to Reviewer 4gc7 (1/3)**
>
> We thank the reviewer 4gc7 for the valuable and positive feedback. We addressed all the comments. Please kindly find the detailed responses below. Any further comments and discussions are welcomed!
>
> **Q1: The difference between Neural Atoms and supernode.**
>
> > The novelty of the proposed method is limited. For example, from the perspective of general graph machine learning, supernodes have been widely used which are the same as the concept of neural atoms.
>
> **Reply:** Thanks for your valuable feedback. Here, we would like to clarify the difference between Nerual Atoms and other supernode methods. Analytically, the pipeline of both supernode and neural atoms for obtaining global graph information can be described as three steps.
>
> - **Information aggregating:** The information of atoms within the molecular graph is aggregated into either supernode or multiple neural atoms with pair-wise connection.
> - **Interaction among node/atoms:** The second step shows differences in interaction among node/atoms. Supernode commonly exists alone, which means there is **only one** super node and thus lacks the ability to interact with others like neural atoms.
> - **Backward projection:** The final step shows differences in terms of the interaction among nodes/atoms. As supernode commonly exists alone, it thereby lacks the ability to interact with others like neural atoms.
>
> Further, we provide a detailed comparison of supernodes and our proposed neural atoms in the following table from five: (1) the number of supernode or neural atoms, (2) the information aggregating strategy, (3) the interaction among them, (4) the method to project the information back to the original graph.
>
> |                                          |                          Super Node                          |                         Neural Atoms                         |
> | :--------------------------------------: | :----------------------------------------------------------: | :----------------------------------------------------------: |
> |             #Atoms / #Nodes              | 1 (In most cases, there is only one single supernode for aggregating the global graph information, as increasing the number of which might not lead to performance improvement.) | $K$ (Our neural atoms can be defined as the proportion of the average number of atoms of the molecular graph dataset, which is **significantly smaller than the original number of atoms. The more neural atoms, the better the performance.**) |
> |         Information aggregating          | The global pooling method, e.g., global sum/mean/max pooling, **treats all the information from nodes the same**, thus lacking diversity among different nodes in the graph. | The multi-head Attention mechanism allows **aggregating information with different weights according to the similarity between specific neural atoms and the original atoms**, which allows diversity among different atoms. |
> | Interaction among supernode/neural atoms | None. (None. (Since supernode usually exists alone, it thus lacks the ability to interact with others.) | A fully connected graph with an attention mechanism to bridge the neural atoms for information exchange. This allows the information located in a different part of the graph, even with a large hop distance, to share information based on their embedding similarities. |
> |           Backward projection            | Direct element-wise adding. The information of the supernode is directly added to the representation of each node, which fuses the information from the single supernode, which might lead to the similarity among different atoms. | Weighted combination. Our proposed neural atoms can fuse the information within according to the similarity score between neural atoms and the atoms in the original molecular graph. Such a mechanism allows the model to obtain diversity representation for further purposes. |
>
> We have added the above comparison to Appendix F; we kindly refer the reviewer to check our latest version of the draft.

---

> > ### Comment · Reviewer_4gc7 · 2023-11-21
> > **Discussions on virtual nodes vs neural atoms**
> >
> > Thanks for the detailed comparison between virtual nodes vs neural atoms. I agree with the differences on interaction and projection. However, the number of virtual nodes could be K (K>1) as well, e.g., representing a subgraph or a community. This is also mentioned by Reviewer 9FRM.

---

> > > ### Author Response · Authors · 2023-11-22
> > > **Further response to Reviewer 4gc7 (2/2)**
> > >
> > > Thus, we could claim that adopting and increasing the number of virtual nodes could not bring a noticeable improvement; by contrast, neural atoms could alleviate these issues and achieve better performance.

---

> > > ### Author Response · Authors · 2023-11-23
> > > **Would you mind checking our responses and confirming whether you have any further questions?**
> > >
> > > Dear Reviewer 4gc7,
> > >
> > > Thanks very much for your time and valuable comments on the number of virtual nodes.
> > >
> > > We understand you might be quite busy. However, the discussion deadline is approaching, and we have only one day left.
> > >
> > > Would you mind checking our further responses and confirming whether you have any further questions?
> > >
> > > Any comments and discussions are welcome!
> > >
> > > Thanks for your attention and best regards.

---

> > > ### Author Response · Authors · 2023-11-23
> > > **[Last-time Reminder] Could you confirm whether our responses have addressed your concern?**
> > >
> > > Dear Reviewer 4gc7,
> > >
> > > As the rebuttal discussion phase ends in **less than 1 hour**, we want to express our gratitude for your engagement thus far. We would like to remind you that after the 22nd (AOE), we will not be able to answer any further questions you may have. We really want to check with you whether our response addresses your concerns during the author-reviewer discussion phase.
> > >
> > > Your feedback is really important to us. We sincerely hope our responses have addressed your concerns and provided satisfactory explanations. Your thoughtful evaluation greatly helps us improve the overall strength of our paper. We sincerely appreciate your dedication and time again.
> > >
> > > Thanks for your attention.
> > >
> > > Authors of #3722

---

> ### Author Response · Authors · 2023-11-20
> **Response to Reviewer 4gc7 (2/3)**
>
> **Q2. About the self-attention mechanism and the running-time comparison.**
>
> > in the Introduction, the authors claimed the disadvantages of transformers, especially the self-attention mechanism, but the way to project atom representations to neural atom representations in the proposed method still uses multi-head attention. Similarly, the controversy also happens in the running time experiment.
> >
> > From the comparison of the running time, there are no clear advantages of the proposed method compared to the transformers which conflicts with the claim in the Introduction (GTs are more computationally expensive than GNNs). What are the reasons? The multi-head attention used in step 1?
>
> **Reply:** Thanks for this comment. Here, we would like to (1) clarify the difference between neural atoms and Graph Transformers w.r.t. the number of atom processing, and (2) the rationality for our neural atoms adopts multi-head attention mechanism as a way of atom grouping strategy. In addition, we provide the (3) explanation for the small running time gaps between GNN enhanced with neural atoms and Transformer-based methods.
>
> (1) Due to the Graph Transformer operating message passing fully connected, it could involve irrelevant information during the message aggregation and update process. In contrast, the neural atom map the atoms in the molecular graph to $K$ neural atoms, which are **far less than the size of the atoms of the original molecular graph**. Such a mechanism allows us to **filter out** potentially irrelevant information, since the attention mechanism aggregates the information according to the similarity between the embedding of neural atoms and the atoms within the original graph.
>
> (2) Rather than processing attention on the fully-connected graph with huge time consumption, the neural atom can reduce the computation burden and thus restrict the running time. Specifically, instead of directly modeling the atom interaction in a fully connected graph, we **map the potential interaction into the space constructed by neural atoms, which is more sparse compared to the original graph** (Table 4/7/8/9, which shows that our method can exceed the baseline with even one-tenth number of node w.r.t. the average number of node in the original graph).
>
> (3) For the running time experiment, the size of the graph (150.94 nodes, 307.3 edges per graph for both LRGB-pepfunc and LRGB-pepstruct) is relatively smaller compared to the graph of product (the Amazon Product-Computer, which have 13,752 nodes with 491,7222 edges in a single graph) or citation networks (the ogbn-arXiv, which have 169,343 nodes with 1,166,243 edges in one single graph). **The inherent nature of the molecular graph we target implies that it would not have a large graph size.** Thus, the running time gap compared to the other Transformer-based method might not be noteworthy.
>
> Alternatively, we provide a running time comparison on the two graph datasets mentioned above, under the same experimental environment mentioned in the draft, to demonstrate the gap between the Transformer-based method and our proposed Nerual Atoms. The statistical information of the two applied datasets and running times are listed in the following tables.
>
> |                         | #Graphs | #Nodes per graph | #Edges per graph |            Task Type            |  Metric  |
> | :---------------------: | :-----: | :--------------: | :--------------: | :-----------------------------: | :------: |
> |       ogbn-arXiv        |    1    |     169,343      |    1,166,243     | Node multi-class classification | Accuracy |
> | Amazon Product-Computer |    1    |      13,752      |     491,7222     | Node multi-class classification | Accuracy |
>
> |       Type       |       Model       |                    ogbn-arXiv                     |               Amazon Product-Computer               |
> | :--------------: | :---------------: | :-----------------------------------------------: | :-------------------------------------------------: |
> |       GNN        |        GCN        | w/ 2.1s (Params. 5352) ; w/o  0.5s (Params. 3032) | w/ 0.2s (Params. 15082) ; w/o  0.2s (Params. 12746) |
> |       GNN        |       GCNII       | w/ 2.2s (Params. 5352) ; w/o  0.4s (Params. 3016) | w/ 0.2s (Params. 15354) ; w/o  0.2s (Params. 12730) |
> |       GNN        |       GINE        | w/ 2.2s (Params. 5624) ; w/o  0.4s (Params. 3304) | w/ 0.3s (Params. 15066) ; w/o  0.2s (Params. 13018) |
> |       GNN        |     GatedGCN      | w/ 2.3s (Params. 6520) ; w/o  0.6s (Params. 4184) | w/ 0.2s (Params. 16234) ; w/o  0.2s (Params. 13898) |
> | Transformer +GNN |        GPS        |                OOM (Params. 6440)                 |                0.4s (Params. 16154)                 |
> |   Transformer    | Transformer+LapPE |                OOM (Params. 5016)                 |                0.4s (Params. 14730)                 |

---

> ### Author Response · Authors · 2023-11-20
> **Response to Reviewer 4gc7 (3/3)**
>
> As shown in the table above, our proposed method **requires limited additional running time** compared to the baseline GNN in both datasets. In contrast, the Transformer approach runs out of memory in ogbn-arXiv with a similar number of parameters. Such results highlight the effectiveness of neural atoms under larger graph scenarios.
>
> **Q3: Experiment on larger datasets and comparisons with the Ewald-based GNN.**
>
> > Experiments have been conducted on only three relatively small-size benchmark datasets. In fact, there are some larger and more recent benchmark datasets such as OC20 and OE62. Representative SOTA methods have not been compared, for instance [1].
> > [1] Ewald-based Long-Range Message Passing for Molecular Graphs, ICML 2023
>
> **Reply:** Thanks for the valuable question! Here, we would like to (1) further clarify the setting of the molecular graph we focus on and (2) the reason why we are not comparing our method with either the OE62 or OC60 datasets. In addition, we provide the (3) explanation for the unavailability of a larger benchmark.
>
> (1) **As we focus on 2D intra-molecule interaction, both OC20 and OE62 3D inter-molecular benchmarks are incompatible with our setting.** These two datasets require the model to have the ability to handle the 3D coordinate information, and the neural atom does not have the ability to handle the 3D coordinate information for now, as we target the 2D scenario. **We have highlighted this point in the revised submission to avoid misunderstanding.**
>
> (2) For the comparison with more recent work like [1], the Ewald-based GNN shares a similar spirit to our paper, but it mainly focuses on the inter-molecule interaction for 3D molecules. **This is completely different from our setting, as we pay more attention to the 2D intra-molecule interaction.**
>
> (3) The scale of the dataset we adopted, i.e., the Long-Range Graph Benchmark, is considered to be larger than the general molecular graph dataset w.r.t. the graph size. We show the comparison of the dataset statistic in the following table:
>
> |                | #Graphs | #Nodes per graph | #Edges per graph |            Task Type             | Metric  |
> | :------------: | :-----: | :--------------: | :--------------: | :------------------------------: | :-----: |
> |  ogbg-molhiv   | 41,127  |       25.5       |       27.5       |   Graph binary classification    | ROC-AUC |
> |  ogbg-molpcba  | 437,929 |       26.0       |       28.1       |   Graph binary classification    |   AP    |
> |  LRGB-pepfunc  | 15,535  |      150.94      |      307.3       | Graph multi-label classification |   AP    |
> | LRGB-pepstruct | 15,535  |      150.94      |      307.3       |   Graph multi-label regression   |   MAE   |
>
> As we are interested in the atom-level interaction in an inductive learning scenario, the OGB link prediction benchmark might not meet our demand as it only provides a single graph with a transductive setting.
>
> Due to the time limitation, we are unable to incorporate the 3D information into the neural atom mechanism, as the 3D coordinate position of the neural atom needs to be carefully initialized and handled to avoid potential bias from the 3D position.
>
> Alternatively, we directly adopt the neural atom to the 3D scenario without 3D coordinate information and provide the comparison with Ewald-based GNN [1] (denoted as Ewald Block) on the OE62[2] dataset. We follow the same setting from [1] and use SchNet[3] as the backbone GNN. Results show that our method achieves **competitive performance even with the absence of 3D coordinate information**, showing the generalization and effectiveness of our proposed neural atom.
>
> |                   | Energy MAE $\downarrow$ | Energy MSE $\downarrow$ | Using 3D information |
> | :---------------: | :---------------------: | :---------------------: | :------------------: |
> | Baseline (SchNet) |         0.1351          |         0.0658          |     $\checkmark$     |
> |   + Ewald Block   |       **0.0811**        |       **0.0301**        |     $\checkmark$     |
> |   + Neural Atom   |       **0.0834**        |       **0.0309**        |       $\times$       |
>
> We have added this discussion to our draft in Appendix D.
>
> [1] Ewald-based Long-Range Message Passing for Molecular Graphs. In ICML, 2023
>
> [2] Atomic structures and orbital energies of 61,489 crystal-forming organic molecules. In Scientific Data, 2020.
>
> [3] SchNet: A continuous-filter convolutional neural network for modeling quantum interactions. In NeurIPS, 2017.

---

> ### Author Response · Authors · 2023-11-21
> **Would you mind checking our responses and confirming whether you have any further questions?**
>
> Dear Reviewer 4gc7,
>
> Thanks very much for your time and valuable comments.
>
> In the rebuttal period, we have provided detailed responses to all your comments and questions point-by-point for the unclear presentations. Specifically, we provided detailed explanations to further clarify
>
> - The difference between Neural Atoms and supernode. (Q1)
> - About the self-attention mechanism and the running-time comparison. (Q2)
> - Experiment on larger datasets and comparisons with the Ewald-based GNN. (Q3)
>
> Would you mind checking our responses and confirming whether you have any further questions?
>
> Any comments and discussions are welcome!
>
> Thanks for your attention and best regards.

---

> > ### Comment · Reviewer_4gc7 · 2023-11-21
> > **Thanks for detailed explainations**
> >
> > I acknowledge the detailed explanations from the authors, especially the newly added experiments. These answers addressed most of my concerns. Therefore, I increase my rating.

---

> ### Author Response · Authors · 2023-11-22
> **Further response to Reviewer 4gc7 (1/2)**
>
> **Thanks for your valuable question!** Here, we would like to clarify the difference between Nerual Atoms and virtual nodes, w.r.t. the performance by increasing their number.
>
> Specifically, we borrow the setting of Tab.3 in our draft and **align the number of neural atoms and virtual nodes** and the backbone GNN they used. We employ the "VirtualNode" data transform from the PyG framework and **set all VNs connected** for a fair comparison. We provide the results in the following table, which is taken on the Peptides-func dataset and evaluated by AP (**the higher, the better**):
>
> | Peptides-func | Method  | #VNs / #NAs = 5 | #VNs / #NAs = 15 | #VNs / #NAs = 75 | #VNs / #NAs = 135 |
> | :----------: | :-----: | :------------------------: | :-------------------------: | :-------------------------: | :--------------------------: |
> |     GCN      |   VNs   |           0.5566           |           0.5543            |           0.5568            |            0.5588            |
> |              | **NAs** |         **0.5962**         |         **0.5859**          |         **0.5903**          |          **0.6220**          |
> |     GINE     |   VNs   |           0.5437           |           0.5500            |           0.5426            |            0.5426            |
> |              | **NAs** |         **0.6107**         |         **0.6128**          |         **0.6147**          |          **0.6154**          |
> |    GCNII     |   VNs   |           0.5086           |           0.5106            |           0.5077            |            0.5083            |
> |              | **NAs** |         **0.6061**         |         **0.5862**          |         **0.5909**          |          **0.5996**          |
> |   GatedGCN   |   VNs   |           0.5810           |           0.5868            |           0.5761            |            0.5810            |
> |              | **NAs** |         **0.6660**         |         **0.6533**          |         **0.6562**          |          **0.6562**          |
>
> We also conducted experiments on the Peptides-struct dataset with the same hyperparameters and evaluated by MAE (**the lower, the better**), shown in the table below:
>
> | Peptides-struct | Method  | #VNs / #NAs = 5 | #VNs / #NAs = 15 | #VNs / #NAs = 75 | #VNs / #NAs = 135 |
> | :------------: | :-----: | :------------------------: | :-------------------------: | :-------------------------: | :--------------------------: |
> |      GCN       |   VNs   |           0.3499           |           0.3492            |           0.3504            |            0.3492            |
> |                | **NAs** |         **0.2635**         |         **0.2581**          |         **0.2575**          |          **0.2582**          |
> |      GINE      |   VNs   |           0.3665           |           0.3614            |           0.3653            |            0.3687            |
> |                | **NAs** |         **0.2624**         |         **0.2565**          |         **0.2580**          |          **0.2598**          |
> |     GCNII      |   VNs   |           0.3686           |           0.3644            |           0.3648            |            0.3632            |
> |                | **NAs** |         **0.2670**         |         **0.2577**          |         **0.2551**          |          **0.2606**          |
> |    GatedGCN    |   VNs   |           0.3425           |           0.3398            |           0.3409            |            0.3374            |
> |                | **NAs** |         **0.2596**         |         **0.2553**          |         **0.2467**          |          **0.2473**          |
>
> As can be seen from the tables:
>
> - **neural atoms achieve consistently and significantly better performance than the virtual nodes** approach, regardless of their number. In both datasets, a larger number of neural atoms could lead to a better performance.
> - in contrast, **virtual nodes achieve almost identical performance with the increase of their number**, even with the pair-wise connections among them.
>
> We speculate that:
> - such a phenomenon shows that **multiple virtual nodes might not learn representative subgraph patterns**, which is crucial for the model to learn long-range interaction.
> - the poor performance of multiple virtual nodes might be caused by the overwhelming aggregated information from all atoms within the graph, which **leads to over-squashing and decreases the quality of the virtual node embeddings**. (Table in Q1, the "Information aggregating.")
> - in addition, the virtual nodes are simply connected to all atoms within the molecular graph without considering their discrepancy. **This could encourage the similarity among atom embeddings, which leads to poor performance.** (Table in Q1, the "Backward projection.")
> - these analyses align with the comparison between the supernode and neural atoms (see Appendix F).

---

### Author Response · Authors · 2023-11-20
**General Response by Authors**

We would like to thank all the reviewers for their valuable comments on our work.

We have received three reviews with positive ratings **3,6,6**.

We are glad that all the reviewers have fair impressions of our work, including
- interesting and important problem (4gc7, 2N34);
- the conducted experiments or visualizations show the effectiveness of the proposed method (4gc7, 2N34);
- theoretical explanations or real-world cases for the effect of the proposed method (9FRM, 2N34);
- generally well-written (4gc7).

**In the rebuttal period, we have provided detailed responses to all the comments and questions point-by-point.** Specifically,
- we further clarify the settings (Q1 for 2N34; Q3 for 2N34;), details (Q2 for 4gc7; Q4 for 9FRM; Q2 for 2N34; Q4 for 2N34; Q4 for 2N34) and contributions (Q1 for 4gc7; Q2 for 9FRM) of our work;
- we add new empirical evaluations with new settings (Q3 for 4gc7;), detailed discussion with new related papers (Q1 for 9FRM), and running time comparison on large-scale datasets (Q2 for 4gc7;);
- we also discussed the possible extension (Q5 and Q6 for 2N34;);
- we amended our draft to address the controversy and mistakes (Q2 for 4gc7; Q3 for 9FRM);
- we have updated our draft with these valuable reviews and highlighted the modifications.

Lastly, we would appreciate all reviewers’ time again. Would you mind checking our response and confirming whether you have any further questions? **We are anticipating your post-rebuttal feedback!**

---

### Meta-Review · Area_Chair_Eyq2 · 2023-12-06

**Metareview:**

This paper introduces Neural Atoms, virtual representations aggregating information from clusters of atoms to help graph neural networks better capture long-range interactions in molecular graphs. This is an important challenge, with modeling such interactions critical for determining chemical properties in drug discovery.

Results across multiple datasets and GNN variants demonstrate Neural Atoms' ability to enhance performance on tasks requiring sensitivity to distant dependencies uncapturable by local message passing. Analyses also reveal the learned groupings align with physically meaningful atomic assemblies.

I recommend accepting this paper spotlighting a simple yet effective application of established concepts to advance an important domain. Addressing comparisons and scalability during revisions will strengthen its presentation and impact.

**Justification For Why Not Higher Score:**

Concerns were raised about the core concept of virtual nodes existing in prior graph representation learning works. Reviewers also critiqued reliance on attention over claimed transformer disadvantages, the lack of exploration on larger graphs or incorporation of 3D positional information, and suboptimal writing quality in later sections. However the authors have addressed most of them to satisfaction during the rebuttal.

**Justification For Why Not Lower Score:**

Reviewers praised the architectural simplicity allowing the technique to augment any GNN, the insight into condensing long paths into single hops, and the thorough experimentation methodology. One reviewer, who expressed the strongest concerns per-rebuttal, was convinced to increase the score from 3 to 6 after an extensive discussion with the authors.

---

### Decision · Program_Chairs · 2024-01-16

Accept (poster)